# Ultrasensitive and high-efficiency screen of *de novo* low-frequency mutations by o2n-seq

Kaile Wang[1,2,3], Shujuan Lai[2], Xiaoxu Yang[4], Tianqi Zhu[5,6], Xuemei Lu[2], Chung-I Wu[2,7,8] & Jue Ruan[1]

Detection of *de novo*, low-frequency mutations is essential for characterizing cancer genomes and heterogeneous cell populations. However, the screening capacity of current ultrasensitive NGS methods is inadequate owing to either low-efficiency read utilization or severe amplification bias. Here, we present o2n-seq, an ultrasensitive and high-efficiency NGS library preparation method for discovering *de novo*, low-frequency mutations. O2n-seq reduces the error rate of NGS to $10^{-5}$–$10^{-8}$. The efficiency of its data usage is about 10–30 times higher than that of barcode-based strategies. For detecting mutations with allele frequency (AF) 1% in 4.6 Mb-sized genome, the sensitivity and specificity of o2n-seq reach to 99% and 98.64%, respectively. For mutations with AF around 0.07% in *phix174*, o2n-seq detects all the mutations with 100% specificity. Moreover, we successfully apply o2n-seq to screen *de novo*, low-frequency mutations in human tumours. O2n-seq will aid to characterize the landscape of somatic mutations in research and clinical settings.

[1] Agricultural Genomics Institute, Chinese Academy of Agricultural Sciences, Pengfei Road No. 7, Dapeng New District, Shenzhen, Guangdong 518120, China. [2] Key Laboratory of Genomics and Precision Medicine, Beijing Institute of Genomics, Chinese Academy of Sciences, Chaoyang, Beijing 100101, China. [3] University of Chinese Academy of Sciences, Shijingshan, Beijing 100049, China. [4] Center for Bioinformatics, State Key Laboratory of Protein and Plant Gene Research, School of Life Sciences, Peking University, Haidian, Beijing 100871, China. [5] Institute of Applied Mathematics, Academy of Mathematics and Systems Science, Chinese Academy of Sciences, Haidian, Beijing 100190, China. [6] Key Laboratory of Random Complex Structures and Data Science, Academy of Mathematics and Systems Science, Chinese Academy of Sciences, Beijing 100190, China. [7] State Key Laboratory of Biocontrol, School of Life Sciences, Sun Yat-Sen University, Guangzhou, Guangdong 510275, China. [8] Department of Ecology and Evolution, University of Chicago, Chicago, Illinois 60637, USA. Correspondence and requests for materials should be addressed to C.-I.W. (email: cw16@uchicago.edu) or to J.R. (email: ruanjue@gmail.com).

Next-generation sequencing (NGS) technologies are dramatically revolutionizing research iacrossthe life sciences. However, the inevitable error rate of NGS approaches, ranging from 0.1 to 1%, remains high and varies among different platforms[1–3] and data processing strategies[3,4]. Unfortunately, cascades of pioneering studies have indicated that somatic mutations (the majority of which are characterized by a low- or ultra-low frequency[5]) play critical roles in the development of tumour heterogeneity[6–8], drug resistance[9,10] and prenatal diagnosis[5,11,12].

Thus, it is particularly urgent, but remains incredibly challenging, to detect de novo, low- or ultra-low-frequency mutations based on NGS platforms. To address this problem, many efforts have been made to develop new and more precise methods[11,13–27]. The majority of these methods utilize unique barcodes (or tags) to eliminate amplification and sequencing errors[5,11,22,24,27]. These methods tag every target molecule with different barcodes. Reads with identical barcodes are regarded as a single 'read family' and are used to perform corrections for amplification and sequencing errors. However, the efficiency of these methods relies heavily on the read number for each 'read family', which leads to tremendous read waste and very low data utilization[23,28–30]. In addition, it is a major challenge to balance the number of DNA fragments used in PCR, the PCR cycle number and the fraction of a sequencing lane, all of which strongly influence the final number of 'read family'[28]. These inherent limitations constrain the application of barcode-based methods, especially for low-frequency mutation analyses of relatively large genomes or genomic regions. An alternative method links different replicates of one original circularized molecule via rolling circle amplification (RCA) in tandem to detect a tag-free 'read family', as is the case for Cir-seq[23,25] or Droplet-CirSeq[29]. Multiple copies of one original molecule in a pair of paired-end (PE) reads constitute one 'read family' and the original molecule can be sequenced multiple times via one PE read by controlling the original DNA fragment size. This method effectively overcomes the disadvantages of barcode-based techniques. However, RCA inevitably introduces severe amplification bias[29,31]. This amplification bias remains a major problem and greatly limits the application for the detection of low-frequency mutations. Thus, sophisticated approaches with highly efficient read usage and low amplification bias are needed for the detection of low-frequency mutations.

In this study, we introduce an innovative method termed o2n-seq, which puts two different copies of one original molecule into a pair of PE reads to eliminate sequencing errors, improve data efficiency and reduce library bias (Fig. 1). O2n-seq combines the advantages of barcode- and RCA-based methods, while overcoming their aforementioned limitations. O2n-seq is able to detect low- and ultralow-frequency mutations with ultralow error rate. According to a systematic evaluation, o2n-seq detects almost all true positive (TP) sites with a 1% mutation frequency among 304 polymorphic sites in mixtures of two Escherichia coli strains and the false positive ratio (FPR) could be decreased to <2% for these sites. Moreover, o2n-seq successfully detects all polymorphic sites in a mixture of two phix174 strains (mixed at a ratio of 1:1,000) with an FPR of 0%. In addition, we demonstrate the application of o2n-seq to discover de novo, low-frequency mutations in the hepatocellular carcinoma (HCC) genome. Finally, we also develop a bioinformatics pipeline for analysing o2n-seq data (Supplementary Software).

## Results

**Library construction and sequencing for o2n-seq.** The library construction and sequencing strategies employed for o2n-seq are schematically illustrated in Fig. 1a. First, genomic DNA (gDNA) is sheared into small pieces that are shorter than the length of a single PE read to ensure that each fragment can be sequenced twice independently in a pair of PE reads (Fig. 1b,c). After end repair and dA-tailing, A-tailed DNA are used to attach a Y-shaped adaptor containing a candidate nicking site (such as dUTP) that can be nicked in the following steps (Fig. 1a, step a5). The DNA sequences with adaptors are then denatured into single-stranded molecules and subsequently circularized by a single-strand DNA ligase. High-fidelity DNA polymerase and primers (either with or without a candidate nicking site or not) can then be used to synthesize the second strand. Next, the USER enzyme is employed to nick the DNA (if the nicking site is dUTP) to generate double-nicked circular DNA. Then, DNA polymerase with strand displacement activity is used in the following step (Fig. 1a, step a6) to perform strand displacement reaction. The amplified DNA then can be easily fit into the standard protocols for NGS library construction.

We sequenced the o2n-seq libraries on the Illumina HiSeq 2,500 platform and generated $2 \times 125$ bp PE reads. The adaptors were first removed from the PE reads (Fig. 1c, blue and orange bars) and low-quality read pairs were filtered out. The consensus sequence (CS) can be determined from Read 1 and Read 2 by aligning them with each other, followed by mapping to the reference genome to accurately identify the variances. Because each copy of target DNA is independently derived from the original molecule, o2n-seq can effectively eliminate PCR errors during library preparation as well as sequencing errors. In addition, the strategy of amplifying every molecule four times before being used to prepare a standard NGS library ensures the uniformity of sequencing read coverage and efficiency of single-nucleotide polymorphism (SNP) calling.

**Data utilization and library bias of o2n-seq.** Two prerequisite factors underlying NGS-based low-frequency mutation detection methods are data efficiency and library bias. However, the data usage efficiency of DNA molecular barcoding strategies is poor[30] (Fig. 2a), it is prohibitively expensive to screen low-frequency mutations in megabase-sized genomes[28]. RCA-based strategies improve the read usage efficiency[23,25,29], but suffer from library bias[29,31] (Fig. 2b).

To evaluate the data efficiency and library bias associated with o2n-seq, we prepared six o2n-seq libraries of phix174 gDNA and generated $\sim 1$ GB of data from each library. We found 39.45% ($\pm 4.22\%$) of the raw reads contained the expected right-on o2n-seq reads. Since one right-on o2n-seq read is considered to be one 'read family', this means that 39.45% of the raw reads are 'read families'. In addition, 100% of all CS, which were determined from the right-on o2n-seq reads, were successfully mapped to the reference genome. Finally, according to the number of base pairs in the raw data and CS data, we calculated that o2n-seq has a data utilization efficiency of 13.65% ($\pm 1.24\%$). This is 30 times higher than that of duplex barcode strategies such as Duplex-seq ($P = 7.59 \times 10^{-6}$, Student's t-test), almost 10 times higher than that of barcode strategies like Safe-SeqS ($P = 1.40 \times 10^{-6}$, Student's t-test) and 2.13 times higher than that of RCA strategies such as Cir-seq ($P = 4.74 \times 10^{-5}$, Student's t-test) (Fig. 2a).

To characterize the library bias of o2n-seq, we first compared the sequencing depth variance with that of RCA- and barcode-based methods. O2n-seq exhibited a read depth coefficient of variance (CV) of 27.59% ($\pm 2.25\%$), which is 3.6 times lower than that of Cir-seq for poliovirus libraries[25] ($P = 7.5 \times 10^{-6}$, Student's t-test), 4.22 times lower than that of Cir-seq for phix174 libraries[29] ($P = 5.3 \times 10^{-3}$, Student's t-test) and 2.5

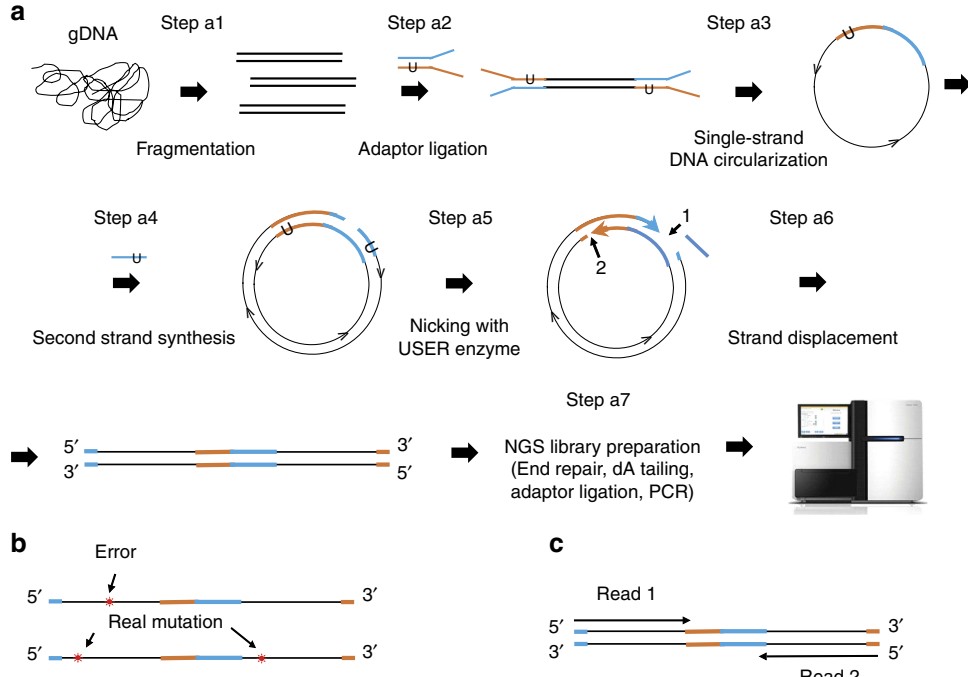

**Figure 1 | Overview of o2n-seq. (a)** O2n-seq workflow. gDNA is sheared into fragments shorter than half the length of the sequencing read (for example, if we sequence reads with PE 125, the length of fragments should shorter than 125 bp) and ligated with Y-shaped adaptors (blue and orange lines) with candidate nicking sites (such as dUTP), then denatured into single-stranded DNA molecules that are circularized using single-stranded DNA ligase. After eliminating linear DNA using DNA exonucleases, the circularized single-stranded DNA is employed for second-strand synthesis. Next, the double-stranded circular DNA is nicked with the USER enzyme and subjected to strand displacement amplification. The purified, strand displacement-amplified DNA can be used to prepare a standard NGS library. The arrows represent the direction of DNA (5′→3′). **(b)** Remove errors. If a variant (red star) is support by only one DNA copy, an error must have occurred at the site, and this type of site is discarded in the following data analysis procedure. However, if a variant is supported by both DNA copies, it is treated as a true variation. **(c)** Using a PE sequencing strategy to sequence o2n-seq reads. Read 1 and Read 2 are sequencing reads of one PE read.

times lower than that of Droplet-CirSeq libraries ($P = 8.6 \times 10^{-4}$, Student's $t$-test) and this CV is comparable to that of Duplex-seq libraries ($P = 0.39$, Student's $t$-test) (Supplementary Fig. 2a). In addition, for the Cir-seq data for poliovirus, the log(e) ratio of the read depth against the average across the whole genome ranged from $-4.59$ ($\pm 0.52$) to 1.94 ($\pm 0.42$) with a median of $-0.34$ ($\pm 0.12$). For the Cir-seq data for *phix174*, the log(e) ratio of the read depth against the average ranged from $-2.42$ ($\pm 0.65$) to 1.91 ($\pm 0.66$) with a median of $-0.43$ ($\pm 0.27$). For Droplet-CirSeq, the log(e) ratio of the read depth against the average ranged from $-2.27$ ($\pm 0.34$) to 1.53 ($\pm 0.45$) with a median of $-0.16$ ($\pm 0.10$). In contrast, for o2n-seq, the log(e) ratio of the read depth against the average ranged from only $-0.68$ ($\pm 0.08$) to 0.63 ($\pm 0.04$), with a median of $-0.03$ ($\pm 0.01$). This indicated that, for this criterion, o2n-seq was also comparable to Duplex-seq and very close to the values associated with the standard NGS method (ranging from $-0.18$ to 0.25 with a median of 0.02) (Fig. 2b and Supplementary Fig. 2b). The depth distribution of every site across the whole genome also indicated that the read depth of o2n-seq was more concentrated around the mean depth (Fig. 2c,g and Supplementary Fig. 2c).

All of these statistics clearly indicated that, compared with barcode-based methods, o2n-seq improved data usage by about 10–30 times, while still displaying comparable library bias. Compared with RCA-based methods, o2n-seq increased the data usage by about two times and greatly improved library bias by displaying four times lower read depth CV. In other words, o2n-seq not only has the advantage of the lower library bias of barcode-based methods, but it also provides higher efficiency data

utilization such as RCA-based methods. These advantages allow o2n-seq to be used to detect low-frequency mutations in megabase-sized genomes and identify mutations more accurately and efficiently.

**Error rate and error pattern of o2n-seq.** To quantitatively evaluate the performance of o2n-seq according to the error rate and error pattern, we first sequenced two disparate *E. coli* strains, *DH5α* and *W3110*, using standard NGS methods. This produced a substantial number of reads with over $300 \times$ coverage for each strain (*W3110*: $335 \times$, *DH5α*: $442 \times$). We screened a total of 375 different sites (Supplementary Data 1) between these two strains. Subsequently, DNA from *DH5α* and *W3110* was mixed at the quantitatively specific ratio of 1:100 to simulate the circumstances of a 1% mutation frequency. The mixture was further sequenced using o2n-seq. For each library, $\sim 100$ million reads were obtained, which were utilized to determine the CS and thus identify variations. For convenience, we defined a variation that was supported by at least one CS as a '1× CSs,' a variation supported by at least two different CSs as a '2× CSs', a variation supported by at least three CSs as a '3× CSs' and so on ('4× CSs' and '5× CSs'). Here, the different types of CSs represent CSs with different sequence contexts (for example, different start points, different lengths or various bases). The error rate of o2n-seq was calculated as the fraction of identified consensus bases that differed from the reference genome beyond the 375 polymorphic sites, which were interpreted as genuine variations rather than errors. Consequently, o2n-seq displayed an error rate of $1.18 \times 10^{-5}$ ($\pm 1.18 \times 10^{-7}$), which is $\sim 100$ times

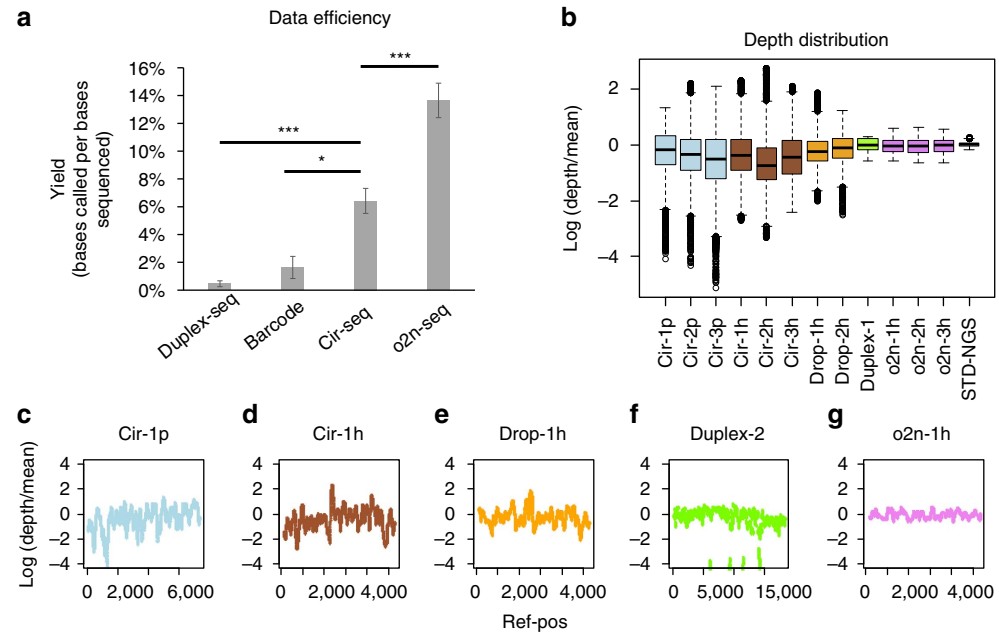

**Figure 2 | Data efficiency and read-depth distribution for various ultrasensitive NGS methods.** (**a**) Data efficiency of Duplex-seq, barcode method (Safe-SeqS), Cir-seq and o2n-seq libraries (means ± s.d.). Two-tailed Student's *t*-test was used for statistical analysis. (**b**) Boxplot of the read depth distribution. The *y* axis represents the log(e) ratio of the depth over the mean for the genome. Three replicates of Cir-seq for poliovirus (Cir-1p, Cir-2p and Cir-3p) (light blue), three replicates of Cir-seq for *phix174* (Cir-1h, Cir-2h and Cir-3h) (brown), two replicates of Droplet-CirSeq (Drop-1h and Drop-2h) (orange), one replicate of Duplex-seq (Duplex-1) (green) and three replicates of o2n-seq (o2n-1h, o2n-2h and o2n-3h) (pink) are plotted. O2n-seq libraries display a more concentrated read depth distribution than the Cir-seq and Droplet-CirSeq libraries and a distribution comparable to Duplex-seq. The depth for each site obtained using o2n-seq is closer to the mean depth value. (**c-g**) Read depth distribution for Cir-seq, Droplet-CirSeq, Duplex-seq and o2n-seq. One example of each type of library is shown here; other cases are shown in Supplementary Fig. 2c. The first and last 100 bp of the genomes are excluded. The Duplex-seq data are from ref. 26, the barcode data are from ref. 11, the Cir-seq data for poliovirus are from ref. 25 and the Cir-seq data for *phix174*, Droplet-CirSeq and STD (standard-NGS, insert size: 90 bp) are from ref. 28.

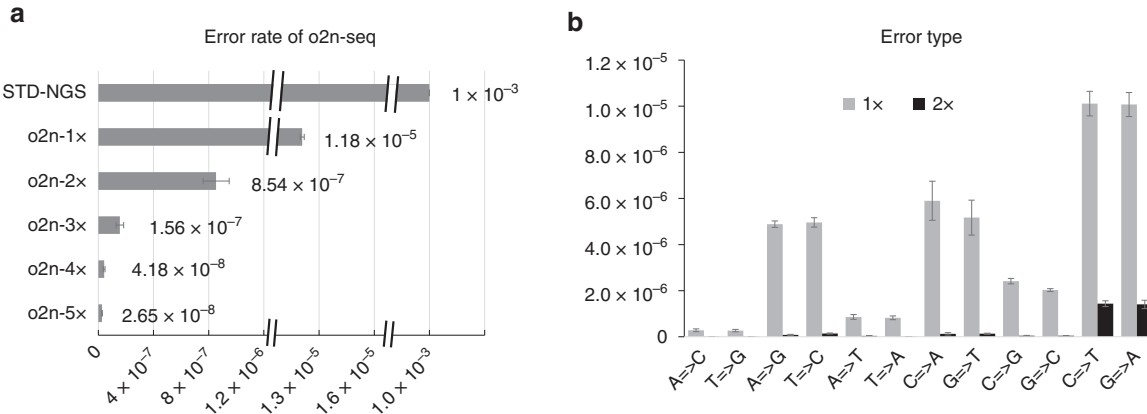

**Figure 3 | Error rates and mutation types obtained by o2n-seq.** (**a**) Error rates of o2n-seq under different CSs support conditions (three experimental replicates, means ± s.d.). STD-NGS, standard next-generation sequencing method. (**b**) Mutation types obtained by o2n-seq for $1\times$ or $2\times$ CSs (three experimental replicates, means ± s.d.).

lower than that of STD-NGS when counting the $1\times$ CSs. The error rate decreased to $8.54\times10^{-7}$ ($\pm9.44\times10^{-8}$), $1.56\times10^{-7}$ ($\pm2.67\times10^{-8}$), $4.18\times10^{-8}$ ($\pm7.03\times10^{-9}$) and $2.65\times10^{-8}$ ($\pm1.76\times10^{-9}$) when counting the $2\times$ CSs, $3\times$ CSs, $4\times$ CSs and $5\times$ CSs, respectively (Fig. 3a). The error rate for $2\times$ CSs was 13.82 times lower than that for $1\times$ CSs. The error rate for $3\times$ CSs was 5.47 times lower than that for $2\times$ CSs. The error rate for $4\times$ CSs was 3.73 times lower than that for $3\times$ CSs. The error rate for $5\times$ CSs was ∼1.58 times lower than that for $4\times$ CSs. These results indicated that the error rate decreases as a site is supported by more CSs. However, the rate of decrease

gradually slows. This may be caused by 'errors' that were actually *bone fide* ultralow-frequency mutations too far from fixation in the *W3110* cell population.

Furthermore, we profiled the pattern of errors for o2n-seq. The error spectrum obtained when counting the $1\times$ CSs indicated that the error rates of two types of transitions ($C=>T$ and $G=>A$) were highest and two other types of transitions ($A=>G$ and $T=>C$) and transversions ($C=>A$ and $G=>T$) exhibited higher error rates than other types of errors (Fig. 3b). As expected, the rates for all types of errors decreased significantly when counting $2\times$ CSs, whereas the error rates of

two types of transitions (C = >T and G = >A) were still highest. After counting 3× CSs, however, only the error rates of eight types that were high for 1× CSs decreased significantly (A = >G,   T = >C,   C = A,   G = >T,   C = >G,   G = >C, C = >T and G = >A). Furthermore, after counting 4× CSs, only error rates of the highest two types (C = >T and G = >A) decreased significantly (Supplementary Fig. 3). Results for 3× CSs and 4× CSs suggested that genuine mutations might still exist in the *W3110* cell population, which could explain the observation that the error rate decreases but the rate of decrease slows when giving more CSs.

**Detecting low- and ultralow-frequency mutations by o2n-seq**. In addition to the error rates and patterns, we evaluated the capacity of o2n-seq to detect low-frequency (1:100) and ultralow-frequency (1:1,000 and 1:10,000) mutations by sequencing two artificially mixed strains of *E. coli* (4.6 Mb) or *phix174* (5.4 Kb). First, to evaluate the performance of screening low-frequency mutations, we sequenced the mixture of DNA from the two *E. coli* strains at a quantitatively specific ratio of 1:100 (*DH5α: W3110*) by o2n-seq (CSs coverage around 800×); for comparison, we also sequenced this mixture and obtained same data size by Cir-seq, which can also be used to screen low-frequency muta-tions in megabase-sized genomes. A set of 304 high-confidence sites, which clearly distinguished the SNPs between the two strains, was considered as the gold-standard in the following analysis (Methods, Supplementary Data 1).

To evaluate the sensitivity of our method for detecting these 1% frequency mutations, we analysed the true mutations detected by o2n-seq compared with that by Cir-seq under different CSs criteria. Results indicated that the number of TP mutations detected by o2n-seq were extremely significant (all $P < 0.001$, Student's *t*-test) more than that of Cir-seq for any CSs criteria (1× − 5×) (Supplementary Fig. 4a). In other words, the sensitivity of o2n-seq was significantly higher than that of Cir-seq. For o2n-seq, 99.12% (±0.69%), 96.49% (±1.00%), 90.46% (±0.33%), 82.24% (±2.61%) and 71.93% (±3.13%) of the gold-standard sites were successfully detected under 1×, 2×, 3×, 4× and 5× CSs criterion, respectively. In contrast, for Cir-seq, only 60.75% (±6.50%), 37.94% (±7.74%), 22.92% (±5.85%), 15.02% (±3.06%) and 10.64% (±2.87%) of the sites were detected under the 1×, 2×, 3×, 4× and 5× CSs criterion, respectively (Fig. 4a). As expected, the number of TP mutations detected by o2n-seq decreased slowly, as more CSs were required to support one mutation.

A good method for mutation detection requires both high sensitivity and a low FPR. To measure the FPR, we comprehen-sively determined the total number of variants detected by o2n-seq versus Cir-seq. The number of variants identified by o2n-seq ranged from 39,021 (±1,150) to 221 (±11), whereas the number found by Cir-seq ranged from 11,913 (±2,265) to 34 (±11) depending upon the CSs criteria (Supplementary Fig. 4b,c). After removing the TP mutations mentioned above, we can see most of the variants detected by o2n-seq and Cir-seq under the 1× CSs and 2× CSs conditions were FP variants (FPR > 60%). However, the FPR decreased to 33.85% for 3× CSs, then sharply decreased to 6.44% (±2.31%) for 4× CSs, to 1.94% (±0.18%) for 5× CSs. Although o2n-seq detected more variants, the FPR of o2n-seq is comparable to Cir-seq (Fig. 4b).

In practice, beyond the error rate and data size, the FP variants could also be distinguished by different characters of errors and real mutations. In general, most of the errors appear randomly but the real mutation does not. We profiled the mutation frequency spectrum of FP and TP variants for o2n-seq under different CSs conditions. As expected, we found the mutation

frequencies of majority of FP variants (99% under 1× CSs, 94% under 2× CSs and 83% under 3× CSs) were lower than 0.005. In contrast, the mutation frequencies of only a very small fraction of TP variants (7% under 1× CSs, 6% under 2× CSs and 4% under 3× CSs) were <0.005 (Fig. 4c and Supplementary Fig. 5). According to the frequency difference, we further filtered the variants detected by o2n-seq, and found the FPRs (under 1× − 3× CSs) after filtering were 1.7–4 times (all $P < 0.001$, Student's *t*-test) lower than unfiltered, whereas the sensitivity slightly decreased. For 4× and 5× CSs, only the FPRs decreased significantly ($P < 0.05$, Student's *t*-test) but the sensitivity did not (Fig. 4a,b) and the FPR of 5× CSs decreased to 1.36% (±0.07%). We could predict that the FPR would be decreased further if more different characters were taken into consideration (such *as priori* knowledge of mutation patterns of different organisms).

Low-bias amplification implies that the allele frequencies of detected mutations should represent the true allele frequencies. As the amplification bias of o2n-seq was verified to be low, the minor allele frequency (MAF) of detected TP mutations in a 1:100 mixture of *E. coli* should approach the theoretical value (0.99%). To validate this, we profiled the MAF of every TP mutation and observed that the MAFs of these mutations ranged from 0.15% (±0.01%) to 3.33% (±0.25%) with a mean value of 1.11% (±0.05%) and a median value of 1.05% (±0.03%). The MAFs of 73.80% (±1.76%) of mutations ranged from 0.5% to 1.5% (Fig. 4d). Based upon these statistics, it can be concluded that the MAF measured by o2n-seq was representative of the true frequency and these results provide further evidence that the library bias of o2n-seq was low.

Next, we assessed the ability of o2n-seq to detect ultralow-frequency mutations. We first applied standard NGS methods to sequence *phix174* DNA from two strains separately, to identify the different sites between these strains. We obtained ~200,000× coverage for each strain. Next, we mixed these two strains at ratios of 1:1,000 and 1:10,000 to prepare o2n-seq libraries and sequenced 4–22 million reads for each library for detecting these ultralow-frequency mutations. Two loci that were heterozygous in one strain but homozygous in the other strain was taken as gold-standard sites (Table 1). For detecting mutations with frequencies 0.07 and 0.08% (1:1,000 mixtures, Table 2), we evaluated the sensitivity and FPR of o2n-seq with regard to CSs number and sequencing coverage using down sampling approach. The results showed that the sensitivity increased as the sequencing depth increased before it reached to 100%; in contrast, the FPR decreased. In addition, the sensitivity decreased as the number of CSs supporting one mutation increased, but the FPR also decreased greatly (Fig. 4e,f and Supplementary Fig. 6). We found o2n-seq has 100% sensitivity and 0% FPR under 6× CSs condition when the depth of CS was 20,000×. Furthermore, the allele frequencies of two heterozygous sites measured by o2n-seq were in accordance with theoretical values (Table 2). We also used o2n-seq to detect mutations with frequencies 0.007 and 0.008%; we found o2n-seq still have 100% sensitivity for these mutations and the allele frequencies of both sites were close to the expected values (Table 2).

**Screening *de novo* low-frequency mutations in HCC tumours**. All of the above experiments showed that o2n-seq greatly reduces the sequencing-related errors and exhibits high efficiency and low bias in screening low-frequency mutations. Next, we employed this method to screen *de novo*, low-frequency mutations in human tumour samples. We separately prepared o2n-seq libraries for one normal sample (N1) and two carcinoma samples (T1 and T2), which were collected from different regions of the

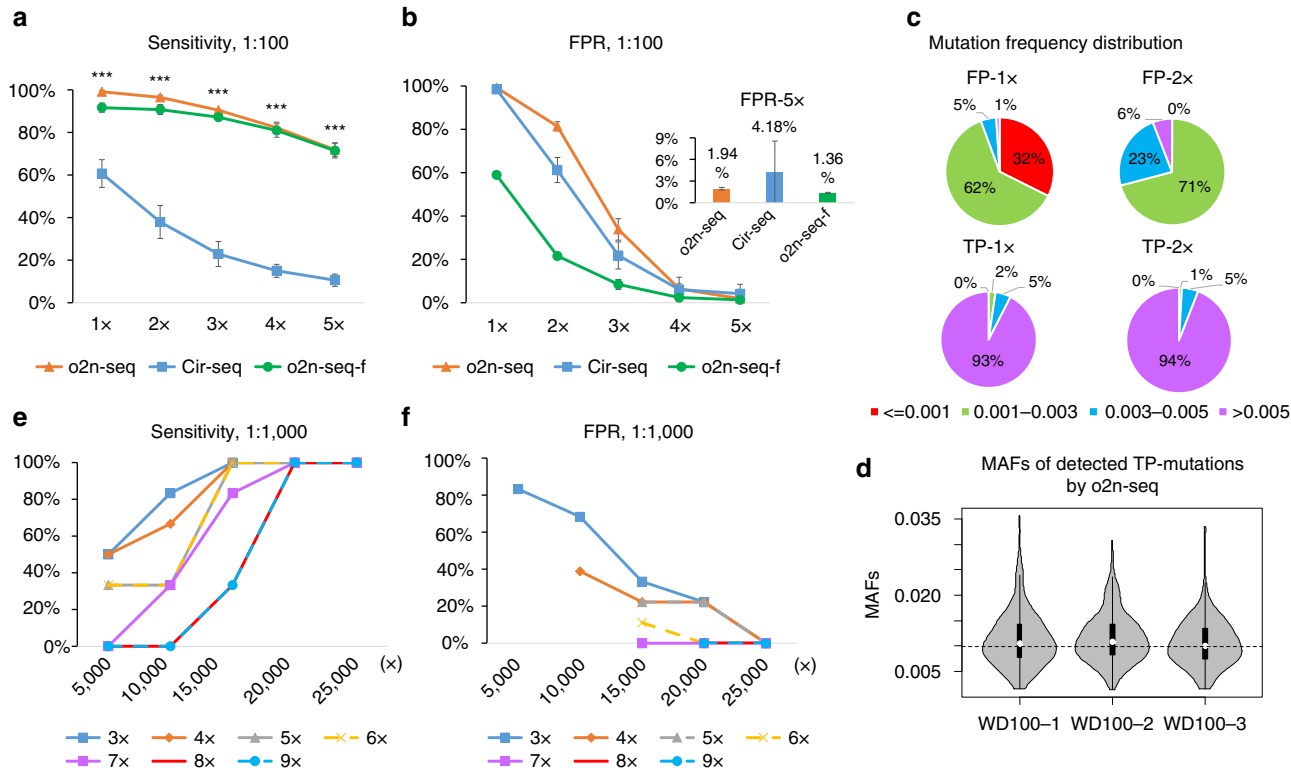

**Figure 4 | Performance of o2n-seq for detecting mutations with 1% and 0.1% allele frequency.** (**a,b**) Sensitivity and FPR of mutation detection of o2n-seq (three experimental replicates, orange), Cir-seq (three experimental replicates, blue) and o2n-seq after filtering with frequency (o2n-seq-f, green) under different CSs criteria for the 1:100 mixture of *E. coli* (means ± s.d.). Two-tailed Student's *t*-test was used for statistical analysis. (**c**) Mutation frequency distribution of FP and TP variants detected by o2n-seq under different CSs (1× and 2×) for the 1:100 mixture of *E. coli*. 3×-5× CSs were showed in Supplementary Fig. 5. (**d**) MAFs of TP mutations detected by o2n-seq for the 1:100 mixture of *E. coli*. The MAFs of three experimental replicates was plotted. The dashed horizontal line indicates the theoretical MAF (0.99%). (**e,f**) Sensitivity and FPR of mutation detection of o2n-seq by different CSs criteria (3× − 9×) under different total CSs coverage (5,000–25,000×) for the 1:1,000 mix of *phix174*. The results of the other experimental replicate were shown in Supplementary Fig. 6. Dash lines were used to display the overlapped results better.

**Table 1 | Allele frequencies of two *phix174* strains (NEB catalog N3021S and Promega catalog D1531).**

| Chr | Pos | Ref | Alt | Promega_AF | NEB_AF |
|-----|-----|-----|-----|------------|--------|
| *phix174* | 3,111 | G | A | 0.3112 | 0.6888 | 1 | 0 |
| *phix174* | 3,133 | C | T | 0.7987 | 0.2013 | 0 | 1 |

**Table 2 | Theoretical and measured allele frequencies of *phix174* mixtures at different ratios.**

| NEB: Promega | $10^3$:1 | | $10^4$:1 | |
|--------------|----------|----------|----------|----------|
| ***phix174*: 3,111** | G | A | G | A |
| Theoretical | 0.99931 | $\mathbf{6.9 \times 10^{-4*}}$ | 0.999931 | $\mathbf{6.9 \times 10^{-5\dagger}}$ |
| Measured | 0.99955 | $\mathbf{4.5 \times 10^{-4*}}$ | 0.999936 | $\mathbf{6.4 \times 10^{-5\dagger}}$ |
| ***phix174*: 3,133** | C | T | C | T |
| Theoretical | $\mathbf{8.0 \times 10^{-4\ddagger}}$ | 0.99920 | $\mathbf{8.0 \times 10^{-5\#}}$ | 0.99992 |
| Measured | $\mathbf{5.2 \times 10^{-4\ddagger}}$ | 0.99948 | $\mathbf{6.0 \times 10^{-5\#}}$ | 0.99994 |

*$P = 0.06$.
†$P = 0.90$.
‡$P = 0.10$.
#$P = 0.50$.
Bold indicates significant *P*-values. (All the *P*-values are larger than 0.05, allele frequencies measured by o2n-seq were in accordance with or close to the theoretical values.)

same section of a human HCC tumour[32] (Supplementary Fig. 7). These libraries were subsequently subjected to capture with a 0.42 Mb target-probe panel (Methods, supplementary Data 2). We obtained 23.5–41.5 million CSs after o2n-seq data processing, which roughly covered the target region by 4,800×, 3,400× and 2,800× for N1, T1 and T2, respectively.

First, we tried to figure out the performance of o2n-seq when combined with target region capture. To evaluate this, as a comparison, we used the same probe panel to capture standard NGS libraries. The results showed, for o2n-seq libraries, 58.40% (±2.35%) CSs data were successfully mapped to the target region and 99.92% (±0.03%) of the total target region was covered, whereas for standard NGS libraries 41.44% (±0.59%) of data were mapped to the target region, and covered 99.95% (±0.05%) target region. These results indicated that the capture efficiency of o2n-seq was comparable with that of standard NGS method.

We then profiled mutations in this target region for each sample. After filtering the mutations in dbSNP and germline mutations, o2n-seq detected 239 and 237 high-frequency somatic

mutations in the T1 and T2 samples, respectively (Supplementary Data 3). The mutation type of those high-frequency mutations (Methods) in tumour samples were identical (Supplementary Fig. 8), but the frequency of those mutations in T1 concentrated on higher frequency than that of T2, indicating the heterogeneity level differed between these two tumours (Supplementary Fig. 9).

Screening of low-frequency mutation was better to demonstrate the substantial advantage of o2n-seq. We detected 4, 2 and 9 low-frequency mutations in N1, T1 and T2 samples, respectively (Supplementary Data 3). The frequency of these mutations ranged from 0.0028 to 0.087 and included different mutation types (Fig. 5a,b). To validate these low-frequency

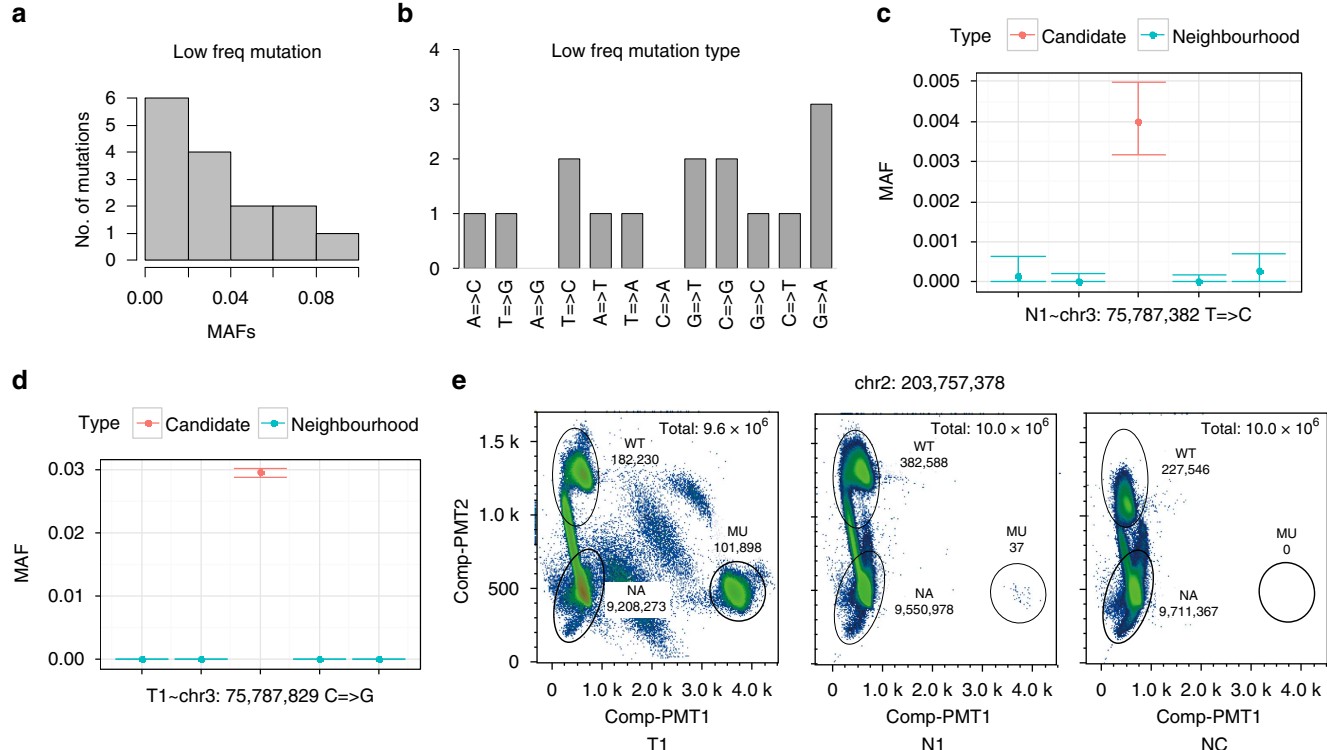

**Figure 5 | Mutation frequency and mutation type distribution for low-frequency mutations.** (**a**) Mutation frequency of 15 low-frequency mutations detected by o2n-seq. The x axis represents the MAF of these mutations (detected by o2n-sam2sites). (**b**) Distribution of mutations type for the 15 low-frequency mutations. (**c,d**) Validated low-frequency mutations by ultra-deep amplicon sequencing (Methods). The x axis represents the mutation (red) and the sequences ± 2 bp from the mutation (blue). The y axis represents the MAF of the mutation. (**e**) Raindrop Digital Droplet PCR was performed to validate ultralow-frequency mutations (~0.1%) in the normal sample. T1 is the positive control (high frequency), N1 is the sample and NC is the negative control. Each dot on the graph corresponds to signal from a single droplet. Comp-PTM2 represents the signal for the wild-type (WT) allele, Comp-PTM1 represents the signal for the mutant allele (MU). NA represents droplets that do not contain any probe targets. Colour of the clusters represents the density of dots.

mutations, we designed ultra-deep amplicon sequencing (from 30,000 × to 250,000 ×) for each candidate. For the mutations that were successfully sequenced by amplicon sequencing, we found that all of the low-frequency mutations in N1, T1 and all but two in the T2 sample, were true polymorphisms (Fig. 5c,d and Supplementary Fig. 10).

Somatic mutations of tumour are expected to be absent in normal control sample, except contamination or metastasis. To perform this test, we investigated whether the high-frequency somatic mutations in tumour samples also existed in the normal sample with ultralow frequency. As somatic mutations in tumour were validated, we used relatively looser data filtering criterion (3 ×) to detected them in N1. We identified two ultralow-frequency mutations (0.12% and 0.16%) in the N1 sample that displayed frequencies of over 20% in both tumour samples (Table 3). Both mutations (100%) were validated by digital droplet PCR successfully (Fig. 5e and Supplementary Fig. 11). However, we noticed the frequencies (measured by digital droplet PCR) of these two mutations were as low as $3.69 \times 10^{-4}$ (chr2: 203,757,378) and $6.84 \times 10^{-4}$ (chr2: 179,247,783), which should not be detected in 4,800 × data coverage under 3 × criterion. It was thought to be sampled up out of hundreds of mutations in the same low frequency level. Predictably, more ultra-low frequency mutations would be detected when the data coverage is deep enough.

## Discussion

The detection of *de novo*, low-frequency mutations is critical for understanding the genetic heterogeneity of cell populations,

**Table 3 | Mutations displaying ultralow frequencies in normal sample (N1) but high frequencies in tumour samples (T1 and T2).**

| Chr | Pos | Ref | Alt | N1 | T1 | T2 |
|-----|-----|-----|-----|----|----|----|
| | | | | **Frequencies** | | |
| chr2 | 179,247,783 | C | T | 0.0012 | 0.40 | 0.36 |
| chr2 | 203,757,378 | T | A | 0.0016 | 0.36 | 0.36 |

locating drug-resistant mutations, studying cancer subclone evolution. To address these problems, we devise an approach termed o2n-seq. This method introduces two strategies to guarantee high efficiency, high sensitivity and low bias for detecting low-frequency mutations. First, two different copies of one original molecule are physically linked in tandem and sequenced separately through a pair of PE reads (for Illumina platforms). This strategy generates one newly formed molecule constituting one 'read family'. This overcomes the low efficiency of read usage in barcode-based library preparation methods (for example, Safe-SeqS and Duplex-seq). Second, the amplification of the original molecule is rigorously constrained by ensuring that each one is amplified only four times during the construction of the tandem molecule. This strategy guarantees that the tandem molecule genuinely represents the state of the original DNA by avoiding over-amplification, which occurs with other methods, such as Cir-seq and Droplet-CirSeq, thus reducing the bias associated with library preparation.

Experiments evaluating different ratios of sample mixtures provided direct evidence for the high sensitivity and low FPR of o2n-seq in detecting low- and ultralow-frequency mutations. This method allows us to more comprehensively characterize the landscape of somatic mutations, particularly those present in a very small fraction of one population. Moreover, the data requirements for o2n-seq are acceptable for detecting ultralow-frequency mutations. These characteristics enable the analysis of low-frequency mutations in species with median-sized genomes (such as *Caenorhabditis elegans* and *Drosophila*) using o2n-seq. In addition, there is still some room for improvement for the sensitivity and data utilization of o2n-seq by optimizing the experimental protocol and data analysis strategies.

To explore its potential utility for discovering *de novo*, low-frequency mutations in human tumour samples, we performed o2n-seq in combination with target region capture to screen mutations in HCC samples. We found 18 *de novo*, low- and ultralow-frequency mutations. In addition, we discovered two high-frequency mutations associated with tumourigenesis existed in the normal samples at an ultralow-frequency of $\sim 0.1\%$ (100% validated). Therefore, o2n-seq provides a more efficient and precise way to study tumour cell evolution, tumour heterogeneity and population genetics.

In addition, o2n-seq exhibits an error rate of $10^{-5}$–$10^{-8}$, which makes it very sensitive for discovering *de novo* mutations and identifying recurrent low- and ultralow- frequency mutations. O2n-seq could be easily adapted for use in different disease diagnosis mutation panels or cancer panels to identify recurrent mutations, screen low-level drug-resistant mutations, discover pathogenic genes or trace low-frequency somatic mutations. We anticipate that our method could be applied in the clinic and personalized precision medicine.

## Methods

**O2n-seq library preparation.** *DNA fragmentation.* gDNA (1–4 μg) was sheared into ∼100 bp fragments in Buffer AE (10 mM Tris-Cl, 0.5 mM EDTA) using Covaris S220 in 130 μl volume (shearing condition: duty cycle: 10%, intensity: 5, cycles per burst: 100, time: 600 s, temperature lower than 4 °C), then purified with Oligo Clean & Concentrator (Zymo Research). The purified DNA was run on a 4% agarose gel at 80 V for 70 min and the gels with DNAs in length of 60–120 bp marked with 20 bp DNA ladder (TaKaRa) were particularly cut off and further extracted using QIAGEN MinElute Gel Extraction Kit (6 × buffer QG). Alternatively, DNA can be sheared with the following shearing conditions: duty cycle: 10%, intensity: 5, cycles per burst: 100, time: 900 s, temperature lower than 4 °C and purified with MinElute Reaction Cleanup Kit without gels cut steps.

*End preparation.* Mix the following components in a sterile nuclease-free tube: End Prep Enzyme Mix (NEBNext Ultra End Repair/dA-Tailing Module, NEB, catalogue number: E7442S) 3.0 μl, end repair reaction buffer (10 ×) 6.5 μl and fragmented DNA 55.5 μl, and place in a thermocycler, with the heated lid on. Next, run the follwing programme: 30 min at 20 °C, 30 min at 65 °C, hold at 4 °C.

*Adaptor ligation.* Add the following components: Blunt/TA Ligase Master Mix (NEBNext Ultra Ligation Module, NEB, catalogue number: E7445S) 15 μl, adaptor (see below) 2 μl, ligation enhancer: 1 μl, directly to the End Prep reaction mixture and mix well, and incubate at 20 °C for 30 min, 65 °C for 5 min in a thermal cycler then put on ice immediately. The mixture was purified with MinElute Reaction Cleanup Kit (QIAGEN). After that, the purified DNA was incubated at 65 °C for another 5 min and immediately put on ice, then purified with 1.8 × AMPure XP beads to further eliminate the adaptor contamination. The ultimate DNA concentration was calibrated using Qubit 2.0 dsDNA HS Assay Kit.

The adapter was synthesized from two oligonucleotides, /5phos/5′-GATCAGT-CGTACGTGCTTACTCTCAATAGCAGTT-3′ and /5phos/5′-GTGGGCAGTC-GGTGAACGACTGAUCT-3′ (Invitrogen, Life Technologies). The adapter strands were annealed by combining equimolar amounts of each oligo to a final concentration of 50 μM in anneal buffer (10 mM Tris-HCl pH 7.5, 1 mM EDTA, 0.1 mM NaCl) and heating to 94 °C for 5 min and then cooling to room temperature by 0.1 °C s$^{-1}$.

*Single-strand DNA circularization.* Thirty microlitres of DNA (<300 ng) was denatured at 95 °C for 3 min and immediately put on ice for another 3 min, then added by a mixture of 3 μl CircLigase II 10 × Reaction Buffer (CircLigase II ssDNA Ligase, Epicentre, catalogue number: CL9021K), 1.5 μl 50 mM MnCl$_2$, 1.5 μl CircLigase II ssDNA Ligase (100 U). The mixtures were further incubated at 60 °C for 14 h, 25 °C for 2 h and the inner enzymes were inactivated by heating at 95 °C for 2 min, and immediately put on ice for another 3 min. Subsequently, 1 μl

Exonuclease I (NEB, M0293S), 1 μl Exonuclease III (NEB, M0206S) and 1 μl Fpg (formamidopyrimidine DNA glycosylase, NEB, M0240S) were added into the reaction and jointly incubated at 37 °C for 1 h. Then the mixture was purified with MinElute Reaction Cleanup Kit (3 × ERC) (QIAGEN) and its final concentration was calibrated with Qubit ssDNA Assay Kit.

*Second-strand synthesis and nicking.* A 15.8 μl sample of circularized DNA, 1 μl of primers and 2 μl of NEBuffer 4 were mixed well and incubated at 95 °C for 3 min, followed by 45 °C for 5 min and immediately placed on ice for another 3 min. Then, 0.5 μl of 2.5 mM dNTPs, 0.2 μl of 100 × BSA and 0.5 μl of T4 DNA Polymerase were added to the mixture, followed by incubation at 25 °C for 30 min and 75 °C for 20 min. Then, 1 μl of USER Enzyme (NEB, M5505S) was added, followed by incubation at 37 °C for 30 min and 50 °C for 4 min, and the sample was immediately placed on ice thereafter. The mixture was finally purified with 1.5 × AMPure XP beads.

*Strand displacement reaction.* The strand displacement of nicked double-stranded circularized DNA was processed in a reaction consisting of 16 μl of nicked DNA, 2 μl of Isothermal Amplification Buffer, 1 μl of 2.5 mM dNTPs and 1 μl of Bst 2.0 WarmStart DNA Polymerase for 1 h at 60 °C. The product was then purified with 1 × AMPure XP beads. The recovered DNA could be used for preparing standard NGS libraries.

**Standard NGS library preparation.** We employed the NEBNext Ultra DNA Library Prep Kit for Illumina (NEB, E7370S) to prepare a standard NGS library, with slight modification. Briefly, end preparation was performed in a reaction containing 18.5 μl of recovered DNA, 2.17 μl of NEBNext End Repair Reaction Buffer and 1 μl of NEBNext End Prep Enzyme Mix, with incubation for 30 min at 20 °C and 30 min at 65 °C. Then, 5 μl of Blunt/TA Ligase Master Mix, 0.33 μl of NEBNext Ligation Enhancer and 0.17 μl of barcode adaptor (Bioo Scientific, NEXTflex DNA Barcodes, 514102) were added to the mixture, followed by incubation at 20 °C for 30 min. The product was purified with 0.8 × AMPure XP beads. PCR was performed in a reaction consisting of 24 μl of ligated DNA, 1 μl of NEXTflex Primer Mix (Bioo Scientific, 514102) and 25 μl of KAPA HiFi HotStart ReadyMix (2 ×) with the following cycling conditions: 98 °C for 45 s and 5–11 cycles of 98 °C for 15 s, 65 °C for 30 s and 72 °C for 1 min, with a final step at 72 °C for 4 min and holding at 4 °C. The product was purified with 0.8 × AMPure XP beads twice or was run on a 2% agarose gel to perform size selection. The purified DNA was then used to perform targeted capture or sequencing.

**Cir-seq library preparation.** Cir-seq libraries were prepared as described previously[23,29] with some modifications. Fragmented DNA was phosphorylated at 37 °C for 30 min in a reaction consisting of 22 μl DNA, 0.5 μl T4 PNK (T4 Polynucleotide Kinase, NEB, M0201S), 2.5 μl T4 DNA Ligase Buffer with 10 mM dATP, then purified with Oligo Clean & Concentrator Kit (Zymo, D4060). Sixteen microlitres of purified DNA was denatured at 95 °C for 3 min followed by incubation on ice for 3 min. Then the sample was supplemented with a mixture of 10 × CircLigase buffer (2 μl), 50 mM MnCl$_2$ (1 μl), CircLigase (1 μl) (Epicentre CL9025K) and further incubated at 60 °C for 14 h before the reaction was stopped by heating at 80 °C for 10 min. Subsequently, 1 μl Exonuclease I (NEB, M0293S) and 1 μl Exonuclease III (NEB, M0206S) were added into the reaction and incubated at 37 °C for 1 h. The enzymes were inactivated at 80 °C for another 20 min. The successfully circularized DNA was purified using the Oligo Clean & Concentrator Kit (Zymo, D4060). The circularized DNA was concentrated to 1 μl, then mixed with 9 μl Sample buffer (illustra GenomiPhi V2 DNA Amplification Kit, GE Healthcare, 25-6600-30). The total 10 μl mixture was denatured at 95 °C for 3 min and left on ice immediately for another 3 min before incubation with 9 μl Reaction Buffer (illustra GenomiPhi V2 DNA Amplification Kit, GE Healthcare, 25-6600-30), 1 μl of Enzyme Mix (illustra GenomiPhi V2 DNA Amplification Kit, NEB, M0280S), 1 μl UDG (Uracil-DNA Glycosylase, NEB, M0280S), 1 μl Fpg (NEB, M0240S) at 30 °C for 35–65 min. The reaction was stopped by incubation at 65 °C for 10 min when the amplification product reached to 0.5–1 μg (monitored by Qubit). The product was then purified with 1 × AMPure XP beads. The recovered DNA (∼500 ng) was sheared into 700 bp, and perform end-repair, dA-tailing using NEBNext Ultra DNA Library Prep Kit for Illumina, then ligated with 1 μl barcode adaptor (Bioo Scientific, NEXTflex DNA Barcodes, 514102). The production was purified with 0.8 × AMPure XP beads and run on a 2% agarose gel to perform the size selection. The gel with DNA in length of 500–800 bp was cut out and further extracted. PCR was performed in a reaction consisting of 24 μl ligated DNA in ddH$_2$O, 1 μl NEXTflex Primer Mix (Bioo Scientific, 514102), 25 μl KAPA HiFi HotStart ReadyMix (2 ×) as the following cycling conditions: 98 °C for 45 s and 8 cycles of 98 °C for 15 s, 65 °C for 30 s, 72 °C for 1 min, then 72 °C for 4 min and held at 4 °C. The production was purified twice with 0.7 × AMPure XP beads. The purified DNA was then used for sequencing (HiSeq 2,500, PE 250).

**Targeted capture.** For o2n-seq human tumour analysis, we chose a patient who was a 75-year-old man with chronic hepatitis B virus infection and liver cirrhosis. The tumour, ∼35 mm in diameter, was on the left lobe of the liver and well encapsulated. It was a histopathological grade III HCC diagnosed at Peking University Cancer Hospital. This study was approved by the Ethics Review Committee

of Peking University Cancer Hospital. Informed consent was signed according to the regulations of the institutional ethics review boards.

We designed a probe panel corresponding to the 687 polymorphic sites of that HCC, from which honeycomb-like microdissected samples were obtained in one plane, which were sequenced using standard NGS methods[32]. This probe panel, total target size: 0.42 Mb (Supplementary Data 2), was customized from Roche NimbleGen. Capture was performed as specified by the manufacturer's instructions, except that one additional customized blocking oligonucleotide (o2n-blocker: 1,000 μM) was added to block the foreign sequence introduced by o2n-seq.

o2n-blocker: 5′-AGATCAGTCGTACGTGCTTACTCTCAATAGCAGCTTG-TGGGCAGTCGGTGAACGACTGATCT-3′.

**AmpliSeq library preparation.** We amplified 100 ng of gDNA from N1, T1, T2 using Ion AmpliSeq Custom DNA Panels (Life Technologies) with the $5\times$ Ion AmpliSeq HiFi Master Mix (99 °C for 2 min and 16 cycles of 99 °C for 15 s, 60 °C for 4 min, then held at 10 °C), respectively. After treatment with 2 μl FuPa reagent (50 °C for 10 min, 55 °C for 10 min and 60 °C for 20 min), PCR products were purified with $1.6\times$ AMPure XP beads. Fupa-treated PCR products were end-repaired and dA-tailed, using NEBNext Ultra DNA Library Prep Kit for Illumina (NEB, E7370S) and then ligated with 1 μl of barcode adaptor (Bioo Scientific, NEXTflex DNA Barcodes, 514102). Adaptor-ligated DNA was cleaned up by dual-size selection, using AMPure XP beads ($0.35\times$ and $1\times$). PCR was performed in a reaction consisting of 30 ng ligated DNA, 1.5 μl NEXTflex Primer Mix (Bioo Scientific, 514102) and 25 μl of KAPA HiFi HotStart ReadyMix ($2\times$) with the following cycling conditions: 98 °C for 45, and 6 cycles of 98 °C for 15 s, 65 °C for 30 s and 72 °C for 1 min, with a final step at 72 °C for 4 min and holding at 4 °C. The product was purified twice with $1\times$ AMPure XP beads. The purified DNA was then used to perform sequencing (HiSeq 4,000, PE 150).

**Digital droplet PCR.** To verify the presence of ultralow-frequency mutations, we separately quantified two mutations on a RainDrop Digital PCR System (RainDance Technologies, Inc.) instrument, as previously described[33]. Customized TaqMan Genotyping Assays, wild-type and mutant assays employed VIC and FAM labels, obtained from Applied Biosystems. gDNA from N1 or T1 were sheared into 3 kb fragments by a Covaris M220 Focused ultrasonicator. A 60 μl PCR reaction was assembled using 1 μg fragmented gDNA, 30 μl TaqMan Genotyping Master Mix ($2\times$, Applied Biosystems), 1.5 μl TaqMan SNP genotyping assays and 2.4 μl stabilizer. The PCR mixture was then transferred into the RainDrop Source emulsion generator (RainDance Technologies, Inc.) to produce emulsified droplets. After emulsion, PCR was performed in an EASTWIN ETC-811 thermo cycler (EASTWIN, Inc. Beijing, China) using the following conditions: 95 °C for 10 min and 40 cycles of 92 °C for 15 s, 60 °C for 1 min, then 98 °C for 20 min and holding at 4 °C with heating and cooling rates of 0.6 °C s$^{-1}$. After amplification, the tube was transferred to the Raindrop Sense machine and the data were analysed with RainDrop Analyst v3 software.

**O2n-seq data processing.** The data processing of o2n-seq reads is different from that for regular re-sequencing data. First, the o2n-adaptor ligated before circularization of the sequence should be removed from PE reads (Supplementary Fig. 1, step 6, highlight with light blue). Second, a CS should be determined from two tandem copies of circular DNA within a pair of PE reads.

Overview of the computational pipeline for processing o2n-seq data.

1. Remove the first five bases and intermediate adaptor from the read pairs (read 1 and read 2).
2. Filter out low-quality read pairs.
3. Determine the CS from read 1 and read 2 by aligning them to each other.
4. Assign quality scores for the CS according to the bases of read 1 and read 2.
5. Map the CS with modified quality to the reference genome and perform variance calling.

*Removal of the adaptor sequence.* Intact o2n-seq reads will consist of 5 bases from the primer used for second-strand synthesis, the target DNA sequence, 62 bases from the combined adaptor and another copy of the target DNA sequence and 2 bases that come from the adaptor after nicking (Supplementary Fig. 1). The bases adjacent to the target DNA sequence were removed. For PE reads, only a very small portion of the intermediate 62 bases of most reads can be sequenced. To obtain the target DNA sequence, the sequenced portion of the intermediate 62 bases and the first 5 bases of read 1 and read 2 were removed.

*Filtering out low-quality read pairs.* Low-quality read pairs were filtered out according to the following principles: (1) bases from the end of a read are cut if they showed a quality score below 20; (2) reads were scanned using a 4four-base-wide sliding window, with cutting when the average quality per base dropped below 20; (3) reads shorter than 36 bases in length were discarded; and (4) reads with an average quality score below 20 were discarded.

*Determination of the CS.* After removing the adaptor sequence and filtering out low-quality reads, the remaining portion of the original reads represented the target DNA sequences. When the target DNA could be sequenced through each single-end read of one PE read, read 1 was the reverse complement to read 2, whereas if the target DNA could not be sequenced through one single-end read, only a portion of the reads were the reverse complement. To determine the CS, read 1 and the reverse complement of read 2 from the same pair of PE reads were aligned to each other, with less than 2 mismatches allowed.

*Assignment of quality scores to the CS.* The quality scores of the CS were based on the original base sequencing quality scores of read 1 and read 2. If one consensus base was supported by both read 1 and read 2, the quality scores of this consensus base were the sum of the base quality scores of read 1 and read 2 (if the sum of the quality scores was larger than '∼', which is the highest score of phred-33, the quality scores of the consensus base were assigned as '∼'). In contrast, if one consensus base was supported by only read 1 or read 2, the quality scores of this consensus base were assigned as '#'.

*Mapping and variance calling.* The CS was mapped onto the reference genome, using BWA[34]. Mapping results containing small insertions and deletions (INDELs), more than 1 mismatch, mapping quality lower than 25 and bases with quality scores lower than 50 were filtered out. The three bases on either side of the CS were trimmed. The remaining bases that differed from the reference genome were treated as variants.

The computational pipeline for processing o2n-seq data and the script to produce '$1\times$ CSs', '$2\times$ CSs',…'n$\times$ CSs' were provided in the Supplementary Software and the following website: https://sourceforge.net/projects/o2n-seq/.

**E. coli and phix174 data processing.** The $2\times125$ PE reads standard NGS data were mapped onto the reference genome using BWA and produce mpileup files using SAMtools[35] (-q 30). SNPs were called using VarScan[36] (v2.3.6). When we analysed the error rate and error pattern of o2n-seq, we ruled out a total of 375 polymorphic sites with a frequency of over 10% detected by VarScan in the E. coli DH5α and W3110 strains from standard NGS data. To evaluate the efficiency of o2n-seq in the detection of low-frequency mutations, we picked up 304 high-confidence sites as the gold-standard from those 375 variants after filtering by total depth ($>100\times$) and the ratio of reference bases to alternative bases ($<0.01$), common SNPs and 4 additional sites (479,519, 479,520, 2,924,565 and 2,924,566). The rest of polymorphic sites (71 sites) along with other low-quality variants existing in the DH5α and W3110 populations (for DH5α, the sites and INDELs with a frequency of $>10\%$ detected by VarScan; for W3110, the sites and INDELs with a frequency of $>1\%$ detected by VarScan (total: 8,109 sites of 4.6 Mb (Supplementary Data 1))) were excluded to eliminate the background noise. To evaluate the efficiency of o2n-seq in the detection of ultralow-frequency mutations, we used two high-confidence different sites detected by VarScan between two phix174 strains as the gold standard. Meanwhile, to minimize the interference of background errors, we excluded two background noise sites (phix174: 1,301, phix174: 1,307), which appeared in all o2n-seq phix174 libraries (unmixed and mixed samples, $n=6$) with frequencies from 0.03% to 0.07%. For detecting variants in 1:1,000 mixtures by o2n-seq, the variants first called by different CSs and then filtered with frequency information (higher than 1/2 theoretical value, 0.0345%).

**Data processing for Cir-seq and Droplet-CirSeq.** The data processing procedures for these two methods were performed essentially as previously described[29,37]. Briefly, a CS was first determined from multiple tandem copies of circular DNA within a sequencing read. Second, the break point of the circularized DNA was detected by mapping it to the reference genome. Third, the CS of the appropriate break point was used for read depth analysis. For E. coli mixture samples, the CS was further processed, as described in the 'Mapping and variance calling' section under 'o2n-seq data processing.'

**Data processing for o2n-seq human capture data.** The procedure was same as that described under 'o2n-seq data processing'. The sorted bam files were further processed using local re-alignment with GATK (v 3.2–2)[38]. The re-alignment bam files were used to produce pileup files, using SAMtools[35] mpileup -q 25 –Q 60 -d 50,000. The re-alignment bam files also were used to produce o2n-pileup files, using o2n-sam2sites. To minimize the FP sites, we employed $8\times$ CSs criteria to identify mutations. The identified mutations were further filtered by VarScan --min-var-freq 0.001 --strand-filter 1 to filter strand bias mutations. These remaining mutations after filtering were treated as candidate polymorphic sites. For low-frequency mutations, the mutation minor allele frequencies calculated by o2n-sam2sites and Varscan should be $<0.1$ concurrently since the MAFs measured by these two different programs were different in linkage variants (one read has at least two variants) and mutational hotspots regions. By the way, the MAFs difference pattern could be used to search for linkage variants and hotspots regions efficiently. Apart from the low-frequency mutations, the remaining mutations were considered as high frequency mutations. High-frequency mutation existent in normal sample were treated as germline mutations.

**Data processing for AmpliSeq data.** The raw reads of AmpliSeq were pre-processed using cutadapt[39] and Trimmomatic[40]. First, for decreasing the occurrence of FPs that occur during the enzymatic steps (Fupa and Illumina adaptor ligation) during NGS library preparation, cutadapt (v 1.10) was used to

trim seven bases from both the 3'- and 5'-end of each read after filtering the Illumina adaptors. Second, Trimmomatic (v 0.33) was employed to filter low-quality reads with default parameters. After preprocessing, data were mapped onto the reference genome (hg 19), using BWA and filtering un-unique mapping reads. To decreases the FP mutations, we then performed local re-alignment of the reads, and base quality score recalibration using GATK (v 3.2–2)[38]. Next, we used SAMtools mpileup (-q 30, -Q 25 and -d 500,000) to generate pileup files.

Pileup files for each locus were filtered with a pileup counting script to separate different bases mapped onto forward and reverse strand. Credible intervals (CIs) were calculate and MAFs were estimated. We measured the strand bias based on GATK FisherStrand after filtering the loci with a depth lower than 500 × . Phred base quality scores of bases supporting major and minor alleles were extracted and measured in a hierarchical Bayesian model described in previous publications[33,41]. Phred base quality scores after realignment and recalibration were first converted into potential sequencing error rate. Distribution of the theoretical minor allele fraction, $\theta$, given the observed minors, $o$, could be estimated with the calculated sequencing error rate, $q$, the total base count, $n$, at the same pileup loci and the unobserved real number of allele count, $r$ as

$$P(\theta|o) \propto P(\theta) \cdot P(o|\theta) = P(\theta) \cdot \sum_r P(o, r|\theta) = P(\theta) \cdot \sum_r P(r|\theta; n)(o|r; q).$$ An

empirical region between $\frac{o}{n} \pm \frac{r}{2\sqrt{n}}$ was uniformly sampled 1,000 times and the likelihood of Bernoulli sampling was calculated as $P(r|\theta; n) = \theta^r (1-\theta)^{(n-r)}$ through an iterative algorithm previously described. Logarithm transformation was used to accelerate the calculation.

Fisher's exact test was carried out for each major_forward, major_reverse, minor_forward and minor_reverse group following GATK recommendation with a hard cutoff of phred transformed Fisher's P-value under 60 (https://software.broadinstitute.org/gatk/documentation/topic?name=methods). The loci that passed this test were used to perform the following validation. For those loci that failed the strand bias, we did find some possible polymorphism sites (Supplementary Fig. 10a,b).

For each validation, ± 2 bp homozygous neighbors, as well as the candidate loci itself, were estimated. If the lower bound of the 95% CI of the candidate loci was higher than any of the upper bound of the 95% CI of the homozygous neighbors, and the upper bound of 95% CI of the candidate loci is under 50%, the candidate site was regarded as positive.

**Data availability.** All raw data were submitted to GenBank with the project accession number of PRJNA339672.

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

## Acknowledgements

We are grateful to Yuxiao Chang for discussions of this manuscript. This work was supported by the Strategic Priority Research Program of the Chinese Academy of Sciences (XDB13040300), National key research and development program of China (2016YFC1200602), Fund of Key Laboratory of Shenzhen (ZDSYS20141118170111640), National Key Basic Research Program of China (2014CB542006), the Natural Science Foundation of China (91531305, 31571353) and Agricultural Science and Technology Innovation Program (ASTIP).

## Author contributions

J.R. and K.W. participated in the design of the study. K.W., S.L. and X.Y. performed the experiments. K.W., J.R., X.Y. and T.Z. performed the bioinformatics analyses. J.R., C.-I.W. and X.L. conceived of the study and participated in its coordination. K.W. and J.R. wrote the manuscript. All authors read and approved the final manuscript.

## Additional information

**Competing interests:** Jue Ruan and Kaile Wang filed a patent application that relates to o2n-seq, the outlined experimental methods and uses thereof. The remaining authors declare no competing financial interests.

