## [Peer Review File · Nature Communications]

Reviewers' Comments:

Reviewer #1 (Remarks to the Author)

The authors present a new sample prep method to improve detection of rare DNA sequences by NGS. The methods looks interesting and useful, but the manuscript is somewhat difficult to follow, and would benefit by further proof-reading. The tumor sequencing application demonstration is not very convincing, so the manuscript is very much a technology proof-of-concept.

Some specific comments:

-Fig 4a-c shows exactly the same data. One way of showing this data is enough. Same goes to the corresponding text in the results section.

-Lines 228-239. In this section it is unclear what cut-off criteria were used for base calling. The cut-off impacts both sensitivity and specificity (false positive rate; FPR). No FPR value is given for the 1:10 000 mix.

-Lines 257-284. This section is the weakest part of the manuscript. A lot of conclusions are drawn from a single tumor section. Here, the study would benefit a lot by increasing the sample size substantially. There are many things that are left unexplained in this section. For example, why should there be high frequency somatic mutations in the normal part of the tissue section? Why does the allele frequency estimates differ so much between the o2n-seq estimates compared to the ultra-deep amplicon sequencing estimates? The observation of traces of tumor mutations in the normal DNA sample might well be due to contamination. The protocol contains 8 reaction steps and 4 purification steps (where of one gel purification step), all potential sources of contamination.

-The authors claim a low bias in amplification and sequencing. One way of proving that, is by presenting the allele frequency of the SNP:s in the normal part of the tissue. This should be 0.5 for all SNP:s. Any deviation from this is caused by bias in amplification and sequencing.

Reviewer #2 (Remarks to the Author)

General comments:

=====

This paper by Wang et al. describes an interesting new sequencing protocol that attempts to reduce error-rates inherent to Illumina's NGS technologies. While roughly similar protocols exist, the method presented here mainly seems to distinguish itself through its more efficient data usage.

The paper is overall nicely written and presented. A few things are unclear though and I have some concerns mainly regarding error rates and the bioinformatics analysis.

I must be fundamentally misunderstanding something, but to me it seems an FP rate of 33% is nothing that you want to advertise in the abstract or elsewhere in fact. From the abstract: "O2n-seq also successfully detects all polymorphic sites (with frequencies of 0.045%-0.052%) in two phix174 mixtures with a false positive rate of

33%." So for every two true negative I get one FP (assuming $FPR = FP/(FP+TN)$)? That would be millions of FPs for any decently sized genome which would render this unusable. I'm surprised by the high error rates for low coverage data for both Cir-seq and o2n-seq (figure 4D). If the reads after correction can be assumed to be mostly correct, how do the authors explain one false positive for every true negative (33% FPR?) at for example 3X where you are supposed to see three "perfect" reads?

The Bioinformatics analysis seems to use a lot of arbitrary filters, which are mostly not motivated .e.g : "Mapping results containing small insertions and deletions (INDELs), more than one mismatch, mapping quality lower than 25, and bases with quality scores lower than 50 were filtered out. The three bases on either side of the consensus sequence were trimmed." What's rationale here?

Along the same lines: why did the authors explicitly not predict indels? I see nothing that would stop them from predicting indels (except the harsh and rather arbitrary filters). Could the authors elaborate?

One of the major selling points of this method is the effective use of data, i.e. less wasted data. With all these filters in place I'd be interested to know what the effective coverage i.e after filtering is.

Minor comments:

=====

* Tumor analysis

This interesting section is unfortunately a bit short (while the other sections are slightly long and dense at the same time).

The fact that Varscan was also run for the tumor is not mentioned in main text even though nice separate validation.

Why did the authors decide not to look at actual somatic mutations, i.e. what about somatic mutations shared between all three tumor samples but not the normal?

There are a few differences between the o2nseq and varscan MAF (see supplementary tables). How can that be explained? Why is the difference highest for normal whereas o2nseq and varscan MAFs tumor samples look much more similar?

* Abstract: "O2n-seq reduces the error rate of NGS to less than 3×10^{-8} per nucleotide sequenced."

This is a bit of an exaggeration, if I understand the results correctly. It's nowhere shown that it's "less" than that rate and in the discussion it's mentioned that the method has an error rate of 10^{-5} to 10^{-8} .

* The abstract doesn't make it clear that this is a new wetlab protocol as opposed to a new sensitive variant calling method.

* Introduction: one not referenced but related wetlab protocol is BaseSeq (PMID 25406369)

* Since the "genomic DNA is sheared into small pieces that are shorter than the length of a single PE read to ensure that each fragment can be sequenced twice independently in a pair of PE reads"

Could the authors show an example length distribution for the actually usable sequence stretch? Even given 2×125 bp PE reads, after removing the adapters and merging the sequences the actually usable fragment must be rather small. Since effectively small, single-end fragments get mapped mapping uncertainty might be at the higher end for larger genomes (e.g. human).

* HCC tumor application

"The sequencing results showed that 59.36% of the total data were successfully mapped to the target region and 99.92% of the total target region was covered. These results are comparable to those of standard NGS capture methods based on this target region panel (40.8% mapped to target region, 100% target region was covered)"

Could the authors provide a reference for this statement? The mappability seems rather low?

* Bioinformatics: unfortunately no scripts or source code were shared. Having said this, the analysis seems simple enough.

* Variant calling

Line 490: "SNPs were called using bcftools view 490 with the options of `-bvcg`". bcftools is the wrong choice of tools here: it assumes diploid organism. Better alternatives are Varscan (as used) and Lofreq or SNVer

* Heterozygosity:

line 495 and table s1: "heterozygous" doesn't apply to E.coli. Might be better to call it "variant site" instead?

* Strand Bias:

line 513: "The identified mutations were further filtered by Varscan --min-var-freq 0.001 to filter strand bias mutations."

That's an MAF filter, not for strand bias filter. But out of interest: did the authors look at strand-bias? This artifact seems to affect many variant calling methods for largely unknown reasons, which are thought to be PCR related. They occur mostly in low-frequency mutations at high coverage. Such a filter is often used (e.g. in samtools) to remove false positives and could potentially help here as well, if it is present at all.

Answer to review #1 reports,

The authors present a new sample prep method to improve detection of rare DNA sequences by NGS. The methods looks interesting and useful, but the manuscript is somewhat difficult to follow, and would benefit by further proof-reading. The tumor sequencing application demonstration is not very convincing, so the manuscript is very much a technology proof-of-concept.

Response: We really appreciate your thoughtful comments. We have updated bioinformatics analysis, performed more experiments and revised the manuscript accordingly. We also addressed the other points you raised. We hope you will be satisfied with the revision. The detailed responses are given below:

Some specific comments:

1 - Fig 4a-c shows exactly the same data. One way of showing this data is enough. Same goes to the corresponding text in the results section.

Response: Thank you very much for your valuable comment. **Figure. 4a-c** were used to show the sensitivity of o2n-seq and Cir-seq. We used the number (**Figure. 4a**), fraction of true positive mutations detected (**Figure. 4b**) and missed (**Figure. 4c**) by those two methods to showed the results. But, as you indicated, there were some redundancy when showed all three figures in the main text. Therefore, we only keep **Figure. 4b** in the main text. We delete **Figure. 4c** and remove **Figure. 4a** into supplementary. Meanwhile, we also delete the redundant text, revise the remaining text accordingly. Thanks a lot.

2 -Lines 228-239. In this section it is unclear what cut-off criteria were used for base calling. The cut-off impacts both sensitivity and specificity (false positive rate; FPR). No FPR value is given for the 1:10 000 mix.

Response: We really appreciate your comments. We evaluated the sensitivity and FPR of o2n-seq with regard to Consensus sequences (CSs) number and sequencing depth using sub-sampling approach for mutations with frequencies between 7×10^{-4} and 8×10^{-4} (1: 1,000 mix). The results showed that the sensitivity increased as the sequencing depth increase before it reached to 100%, in contrast, the FPR decreased. In addition, the sensitivity decreased as the number of CSs supporting one mutation increased, but the FPR also decreased greatly (**Figure. 4e-f, Supplementary Figure. 6**). We found o2n-seq has 100% sensitivity and 0% FPR under 6X CSs condition when the sequencing depth was 20,000 ×. For mutations with frequencies between 7×10^{-5} and 8×10^{-5} (1: 10,000 mix), we found o2n-seq have 100% sensitivity for these mutations and the allele frequencies of both sites were close to the expected values, but the FPR was very high (> 90%) even under the 6X CSs criteria. One part of these FP sites might be some genuine variants, which exist in *phix174* (10,000) with ultra-low frequencies that failed to be detected by standard NGS method, but they were heterozygous sites,

and detected by o2n-seq. The other part were the real false positive sites. We think it might be more appropriate to use o2n-seq to detect recurrent mutations rather than *de novo* mutations if the frequencies of these mutations are such low (8×10^{-5}).

We really appreciate for the helpful suggestions and have included these results in the revised manuscript.

Figure 4 (e-f) Sensitivity and FPR of mutation detection of o2n-seq by different CSs (3X-9X) criteria under different total CSs coverage (5,000-25,000 x) for the 1: 1,000 mix of *phix174*. The results of the other experimental replicate were showed in **Supplementary Fig. 6**. Dash lines were used to display the overlapped results better.

Supplementary Figure 6. Sensitivity and FPR of mutation detection of o2n-seq by different CSs criteria (3X-9X) under different total CSs coverage (5,000-25,000 x) for the 1: 1,000 mix of *phix174* (the other experimental replicate). Dash lines were used to display the overlapped results.

3 - Lines 257-284. This section is the weakest part of the manuscript. A lot of conclusions are drawn from a single tumor section. Here, the study would benefit a lot by increasing the sample size substantially. There are many things that are left unexplained in this section. For example, why should there be high frequency somatic mutations in the normal part of the tissue section? Why does the allele frequency estimates differ so much between the o2n-seq estimates compared to the ultra-deep amplicon sequencing estimates? The observation of traces of tumor mutations in the

normal DNA sample might well be due to contamination. The protocol contains 8 reaction steps and 4 purification steps (where of one gel purification step), all potential sources of contamination.

Response: We appreciate your comments. In fact, there are no high frequency somatic mutations existed in normal tissue. In the main text, we described “After filtering the mutations in dbSNP, we detected 29, 263, and 268 mutations in the N1, T1, and T2 samples, respectively.” In this case, although we filtered the mutation in dbSNP, we didn’t filter the germline mutation of this patient. The reason why we did this is we tried to find out whether there were some high frequency mutations in tumor samples also existed in normal samples but with low frequency. This kind of information would be helpful to infer the tumor evolution fingerprints. However, we also felt it was confusing as you pointed. In the main text, we revised this paragraph into “We then profiled mutations in this target region for each sample. After filtering the mutations in dbSNP and germline mutations, we detected 239, and 237 point mutations in the T1 and T2 samples, respectively (**Supplementary Table S3-S5**). The mutation type of high-frequency somatic mutations in tumor samples were identical (**Supplementary Figure. 8**), but the frequency of somatic mutations in T1 concentrated on higher frequency than that of T2, indicating the heterogeneity level differed among these two tumors (**Supplementary Figure. 9**).” to make the points we tried to expressed more clearly.

The minor allele frequency (MAF) differences between o2n-seq and amplicon-seq might be caused by two major reasons. 1) Reads filtering in o2n-seq (discovering) is stricter than in amplicon-seq (validating). In amplicon-seq, reads that used to produce pileup files only are filtered by mapping quality and base quality before used to call variants and calculated the MAFs; In contrast, CSs of o2n-seq that used to produce o2n-pileup are filtered not only by the mapping and base quality, but also by mismatch number (<2), INDELS et.al (see section “Mapping and variance calling” of “o2n-seq data processing”). CSs covering linkage variants (at least two variants existed in one CS) will be filtered before calculating frequency. So, the MAFs measured by o2n-sam2sites would be underestimated after discarding the CSs with more than one variants. For example, the variant chr3: 75,787,829, whose neighborhood sites (\pm 50bp): chr3: 75,787,794 and chr3: 75,787,809 are also variants (**Supplementary Table S3-5**), in this region, most of CSs contain at least two variants, and these CSs which contain at least two variants would be discarded by o2n-sam2site (script that used to measure MAFs in o2n-seq) before calculating MAFs, so the estimated MAFs of variant chr3: 75,787,829 by o2n-seq are very low. In contrast, the estimated MAFs of this variant by amplicon-seq are higher (**Table r1**). 2) The approaches of enriching target regions for o2n-seq and amplicon-seq are capturing with a target-probe panel and PCR amplification using target specific primers respectively. Previous study¹ showed the PCR amplification of amplicon-seq could produce library bias, and the bias would further affect MAFs. In addition, the capture procedure could also introduce the library bias, and the bias result in MAFs differences. In the response of your next questions (4 - The authors claim a low bias in ...), we provide the evidences that capture could affect the library bias (see below).

	chr	pos	ref	alt	MAF (o2n-sam2sites)	MAF (Amplicon-seq)	Fold (Amp/o2n)
T1	chr3	75,787,829	C	G	0.0084563	0.03	3.547638
T2	chr1	11,009,799	T	A	0.0717241	0.3	4.182695
	chr3	75,787,829	C	G	0.0029286	0.035	11.95114

Table r1 Validated mutations whose MAFs differ much measured by o2n-seq (o2n-sam2sites) and Amplicon-seq.

Yes, the observation of traces of tumor mutations in the normal DNA sample might be due to contamination too. The conclusion of metastasis holds only in the exclusion of DNA contamination. After checking it step by step, we thought that if both mutations were contamination, they only can be contaminated in the genomic DNA (gDNA) extraction step, not the o2n-seq library preparation step, because the DNA used to perform digital droplet PCR validation were the original gDNA not o2n-seq library DNA. To say the least, such kind of contamination can be avoided easily through extracting the gDNA separately. In this section, we demonstrated o2n-seq detected two ultralow-frequency variants and both were validated successfully. We tried to show the high sensitivity and low FPR of o2n-seq in detecting low frequency mutations in tumor samples, but not paid much attention with the biology of tumor evolution. Now, we weaken the previous conclusion into “Somatic mutations of tumor are expected to be absent in normal control sample excepting contamination or metastasis. To perform this test, we investigated whether the high-frequency somatic mutations in tumor samples also existed in the normal sample with ultralow-frequency. As somatic mutations in tumor were validated, we used relatively looser data filtering criterion (3X) to detected them in N1. We identified two ultralow-frequency mutations (0.12%, 0.16%) in the N1 sample that displayed frequencies of over 20% in both tumor samples (**Table 2**). Both mutations (100%) were validated by digital droplet PCR (ddPCR) successfully (**Fig. 5e, Supplementary Fig. 11**).”

4 - The authors claim a low bias in amplification and sequencing. One way of proving that, is by presenting the allele frequency of the SNP:s in the normal part of the tissue. This should be 0.5 for all SNP:s. Any deviation from this is caused by bias in amplification and sequencing.

Response: We appreciate your comments. It’s really a good way to measure allele bias. In the main text, we concluded o2n-seq exhibited low amplification and sequencing bias, which was comparable with standard NGS method. To prove this conclusion, we showed the MAF distribution of the *E.coli* mixtures (1: 100, **Figure. 4d**) and MAF distribution of *phix174* mixtures (1: 1,000 and 1: 10,000, **Table 1**) measured by o2n-seq. As the results indicated, the MAFs measured by o2n-seq were in line with the expected MAFs in all three mixtures (1: 100, 1: 1,000, 1: 10,000).

The allele frequency of the SNP:s in the normal part of the tissue should be around 0.5 if the bias of o2n-seq was low. Please note that, the o2n-seq library of normal part of tissue was subjected to capture with a 0.42 Mb target-probe panel, which was another factor influencing the allele frequency²⁻⁴. The allele frequency

distribution of this sample was showed in **Figure. r1a**. In this figure, the allele frequencies of most of SNP:s were concentrated on 0.5. However, we also observed the allele frequencies of a small fraction sites lower than 0.3. To test allele frequency deviation came from the bias of o2n-seq or targeted capture, we used the same 0.42 Mb target-probe panel to capture two additional standard NGS libraries (not o2n-seq library), the input of whom were normal human blood DNA. We found the allele frequencies distribution of these two standard NGS libraries was in line with that of o2n-seq (**Figure. r1b, c**), the distribution of library “STD-normal-2” was even worse than o2n-seq. This strongly suggested the capture bias. All of results clearly indicated the allele frequency deviation mainly came from the targeted capture not library preparation methods (o2n-seq or standard NGS), which suggested the bias of o2n-seq was comparable with that of standard NGS method whose bias was acceptable.

Figure 4 (d) MAF of TP mutations detected by o2n-seq. The MAF of three experimental replicates was plotted. The dashed horizontal line indicates the theoretical MAF (0.99%) of samples of a 1:100 mixture of *E. coli*.

Table 1 | Allele frequencies of two *phix174* strains (NEB, catalog: N3021S, and Promega, catalog: D1531) (**a**) and theoretical and measured allele frequencies of mixtures at different ratios (**b**).

a

Chr	Pos	Ref	Alt	Promega_AF		NEB_AF	
phix174	3,111	G	A	0.3112	0.6888	1	0
phix174	3,133	C	T	0.7987	0.2013	0	1

b

NEB: Promega	$10^3: 1$		$10^4: 1$	
phix174: 3,111	G	A	G	A
Theoretical	0.99931	6.9×10^{-4}	0.99931	6.9×10^{-5}
Measured	0.99955	4.5×10^{-4}	0.99936	6.4×10^{-5}
phix174: 3,133	C	T	C	T
Theoretical	8.0×10^{-4}	0.99920	8.0×10^{-5}	0.99992
Measured	5.2×10^{-4}	0.99948	6.0×10^{-5}	0.99994

Figure r1 Allele frequency distribution of the SNP:s. **(a)** The frequency distribution of SNP:s measured by o2n-seq in HCC15-N1 (normal part of the tissue). **(b, c)** The frequency distribution of SNP:s measured by standard NGS method in blood samples from two different normal individuals, the SNP:s were called using VarScan. All these three libraries (one o2n-seq library and two standard NGS libraries) were subjected to captured with the same target-probe panel (0.42Mb) before sequencing.

Reference for the responses to reviewer #1

1. Melanie, S. *et.al.* Insight into biases and sequencing errors for amplicon sequencing with the Illumina MiSeq platform. *Nucleic Acids Res.* 2015 Mar 31; 43(6): e37)
2. Samorodnitsky, E. *et al.* Comparison of Custom Capture for Targeted Next-Generation DNA Sequencing. *The Journal of Molecular Diagnostics* 17, 64-75 (2015).
3. Schmitt, M.W. *et al.* Sequencing small genomic targets with high efficiency and extreme accuracy. *Nat Methods* 12, 423-5 (2015).
4. Chung, J. *et al.* The minimal amount of starting DNA for Agilent's hybrid capture-based targeted massively parallel sequencing. *Scientific Reports* 6, 26732 (2016).

Answer to review #2 reports,

General comments:

=====

This paper by Wang et al. describes an interesting new sequencing protocol that attempts to reduce error-rates inherent to Illumina's NGS technologies. While roughly similar protocols exist, the method presented here mainly seems to distinguish itself through its more efficient data usage. The paper is overall nicely written and presented. A few things are unclear though and I have some concerns mainly regarding error rates and the bioinformatics analysis.

1. I must be fundamentally misunderstanding something, but to me it seems an FPR of 33% is nothing that you want to advertise in the abstract or elsewhere in fact. From the abstract: "O2n-seq also successfully detects all polymorphic sites (with frequencies of 0.045%-0.052%) in two phix174 mixtures with a false positive rate of 33%." So for every two true negative I get one FP (assuming $FPR = FP/(FP+TN)$)? That would be millions of FPs for any decently sized genome which would render this unusable. I'm surprised by the high error rates for low coverage data for both Cir-seq and o2n-seq (figure 4D). If the reads after correction can be assumed to be mostly correct, how do the authors explain one false positive for every true negative (33% FPR?) at for example 3X where you are supposed to see three "perfect" reads?

Response: We really appreciate your suggestions and comments. In theory, the number of FP variants are the product of error rate and data size. For example, if we sequence a 5 Mb sized haploid genome with 1,000 × coverage, and this sequencing method has an error rate of 1×10^{-5} . It will produce $(5 \times 10^6) \times 1000 \times (1 \times 10^{-5}) = 5 \times 10^4$ errors. Meanwhile, if the genome has 300 true mutations, so the $FPR = \text{number FP variants} / (\text{number of FP variants} + \text{true mutations}) = 5 \times 10^4 / (5 \times 10^4 + 300) = 99.4\%$. In practice, however, beyond the error rate and data size, most of FP variants could also be distinguished by the different characters of errors and real mutations. In the 1:100 *E.coli* mix, we profiled the mutation frequency spectrum of FP and TP variants for o2n-seq under different CSs conditions. We found the mutation frequencies of majority of FP variants (99% under 1X CSs, 94% under 2X CSs and 83% under 3X CSs) were lower than 0.005. In contrast, the mutation frequencies of only a very small fraction of TP variants (7% under 1X CSs, 6% under 2X CSs and 4% under 3X CSs) were lower than 0.005 (**Figure. 4c, Supplementary Figure. 5**). According to the frequency difference, we further filtered the variants detected by o2n-seq, and found the FPRs (under 1X-3X CSs) after filtering were 1.7-4 times (all $p < 0.001$) lower than unfiltered, whilst the sensitivity slightly decreased. For 4X and 5X CSs, only the FPRs decreased significantly ($p < 0.05$), but sensitivity didn't (**Figure. 4a-b**), and the FPR of 5X CSs decreased to 1.36% ($\pm 0.07\%$). We could predict that the FPR would be decreased further if more different characters were taken into consideration (such as priori knowledge of mutation patterns of different organisms).

For mutations with frequencies between 7×10^{-4} and 8×10^{-4} (*phix174* 1: 1,000

mix), we reevaluated the sensitivity and FPR of o2n-seq with regard to CSs number and sequencing depth using sub-sampling approach. We also took the frequency information into consideration (the frequencies of variants larger than 0.0345% (1/2 theoretical value) were retained). The results showed that the sensitivity increased as the sequencing depth increasing before it reached to 100%, in contrast, the FPR decreased. In addition, the sensitivity decreased as the number of CSs supporting one mutation increased, but the FPR also decreased greatly (**Figure. 4e-f, Supplementary Figure. 6**). We found o2n-seq has 100% sensitivity and 0% FPR under 6X CSs condition when the sequencing depth was 20,000 × for detecting those mutations whose frequencies were lower than 1×10^{-3} .

We really appreciate for your helpful suggestions and have included these results in the revised manuscript and also revised the abstract accordingly.

Figure 4 | Sensitivity, false positive ratio (FPR) and minor allele frequency (MAF) obtained by o2n-seq to detect mutations with 1% and 0.1% allele frequency. **(a-b)** Sensitivity and FPR of mutation detection of o2n-seq, Cir-seq, and o2n-seq after filtering with frequency (o2n-seq-f) under different CSs criteria for the 1:100 mixture of *E.coli*. **(c)** Mutation frequency distribution of FP and TP variants detected by o2n-seq under different CSs (1X, 2X) for the 1:100 mixture of *E.coli*. 3X-5X CSs were showed in **Supplementary Fig. 5**. **(d)** MAFs of TP mutations detected by o2n-seq for the 1:100 mixture of *E.coli*. The MAFs of three experimental replicates was plotted. The dashed horizontal line indicates the theoretical MAF (0.99%). **(e-f)** Sensitivity and FPR of mutation detection of o2n-seq by different CSs criteria (3X-9X) under different total CSs coverage (5,000-25,000 ×) for the 1:1,000 mix of *phix174*. The results of the other experimental replicate were showed in **Supplementary Fig. 6**. Dash lines were used to display the overlapped results better.

Supplementary Figure 5. Mutation frequency distribution of FP and TP variants detected by o2n-seq under different CSs (3X-5X) for the 1:100 mixture of *E.coli*.

Supplementary Figure 6. Sensitivity and FPR of mutation detection of o2n-seq by different CSs criteria under different total CSs coverage for the 1:1000 mixture of *phix174* (the other experimental replicate). Dash lines were used to display the overlapped results.

2. The Bioinformatics analysis seems to use a lot of arbitrary filters, which are mostly not motivated .e.g : "Mapping results containing small insertions and deletions (INDELs), more than one mismatch, mapping quality lower than 25, and bases with quality scores lower than 50 were filtered out. The three bases on either side of the consensus sequence were trimmed." What's rationale here?

Response: We appreciate your comment. There are two classes of filters in our analysis. The first class is to eliminate mapping error, the other is to eliminate low quality bases. Ideally, o2n-seq can detect low frequency INDELS in cell populations too. However, INDELS are tended to make the alignment ambiguous. That means we might call wrong mutations (INDELS and SNPs) around INDELS. We know that re-alignment will improve the accuracy of gaped alignment, but it is still uncertain when comes to detect ultra-low frequency mutations. Furthermore, novel sequences (absent from reference) in population will be a big source of mapping errors, which will increase the number of mismatches. Hence, we filtered read alignments with “small insertions and deletions (INDELS), more than one mismatch, mapping quality lower than 25”. Secondly, in the section of “o2n-seq data processing”, we described the principle of reassigning quality scores to the consensus sequence, for consensus base the quality scores of which were the sum of the base quality scores of read 1 and read 2. “The bases with quality scores lower than 50” implied that the bases quality of at least one original bases from read 1 or read 2 was lower than 25, which was low quality and low confident, and will increase the false positive variants. At last, as showed in **Figure. 1c**, two sides of original DNA fragments were adapter sequences. Previous studies¹⁻⁴ indicated the possibility of occurrence of false positives in the junction of ligation would increase, which may be caused during the enzymatic steps. So, we used the filter “the three bases on either side of the consensus sequence were trimmed” to decrease the false positive variants. In a word, the purpose of all these harsh filters is to minimize the false positive variants during data analysis. The same filters were also used in our previous study^{3,4}.

Figure 1 (c) Using a paired-end sequencing strategy to sequence o2n-seq reads. Read 1 and Read 2 are sequencing reads of one PE read. The blue and brown colors were adapters.

3. Along the same lines: why did the authors explicitly not predict indels? I see nothing that would stop them from predicting indels (except the harsh and rather arbitrary filters). Could the authors elaborate?

Response: We really appreciate your comments. We think it is an open question to be studied in coming work. The reason why we explicitly not predict INDELS is to minimize the mapping error. As mentioned in the response to your second question (the first response), in the main text, we sequenced the o2n-seq libraries using HiSeq

2500 PE125 module, and the length of CSs concentrated on 80~120 bp (**Figure. r2**). Besides that, INDELs may bring ambiguous alignment, short read length often implies high error rate of misplacing, especially when referring to the complex or large genomes. In this situation, if we still allow gaps in mapping procedure, the mapping error will further increase and will lead to many false positive calls. However, if we prepare and sequence the o2n-seq libraries using longer read length platform or modules, such as HiSeq PE250 or PE300 modules, in which the length of CSs for o2n-seq can go up to 250bp or 300bp, or the sequenced genome is very simple, and without complex regions, o2n-seq could be used to predict INDELs. We can imagine that there are still huge bioinformatics works to make the data from o2n-seq couple with calling of ultra-low frequency INDELs. In this study, we focus more on providing a novel experimental technology, but leaving space for further utility of it by any researchers.

Figure r2 Length of CSs distribution. The length distribution of CSs for three experimental replicates of *E.coli* mixture (1:100) was plotted. 1,000,000 CSs of each replicate were used to plot.

4. One of the major selling points of this method is the effective use of data, i.e. less wasted data. With all these filters in place I'd be interested to know what the effective coverage i.e after filtering is.

Response: Thank you for your comments. We calculated the data utilization effective after filtering the data with all those filters (INDELs, mismatch > 1, mapping quality < 25, bases quality < 50, and 3 bases on both sides of CSs), we found 89.33% ($\pm 0.47\%$, n = 6) bases of CSs were remained. If using the filtered CSs data, o2n-seq has the data utilization efficiency of 12.19% ($\pm 1.13\%$), which is still very significantly higher than other methods (27 times higher than Duplex-seq ($p = 8.78 \times 10^{-6}$), 7 times higher than Safe-seqS ($p = 1.98 \times 10^{-6}$), 2 times higher than Cir-seq ($p = 1.24 \times 10^{-4}$)) (**Figure. r3**).

Figure r3 Data efficiency of various ultrasensitive NGS methods. The data efficiencies of Duplex-seq, Barcode, Cir-seq, and o2n-seq were calculated based on the CSs ((bases of CSs) / (bases of raw data)). The data efficiency of o2n-seq-filtered were calculated based on the harsh filtered CSs ((bases of CSs after filtering) / (bases of raw data)).

Minor comments:

=====

* Tumor analysis

This interesting section is unfortunately a bit short (while the other sections are slightly long and dense at the same time). The fact that VarScan was also run for the tumor is not mentioned in main text even though nice separate validation.

Response: We appreciate your suggestions. In the tumor section, we added a paragraph to show more evidences that o2n-seq is compatible with target region capture. In this main text, we aim to introduce the o2n-seq method, and prove its high data efficiency and high accurate. We provided the examples to prove o2n-seq can be used to detect low-frequency mutations in captured tumor samples and those low-frequency mutations were high accurate and can be validated. As for the function of these mutations and the roles they played in the cancer cell evolution, it would be further investigated in detail in our future studies. In addition, we go through the manuscript and delete some redundant statements (highlighted in the main text), especially in the section of “Evaluation of the efficiency of detecting low- and ultralow-frequency mutations by o2n-seq”. Previous **Figure. 4a-c** were used to show the sensitivity of o2n-seq and Cir-seq, just in different ways. To illustrate those result concisely, we only kept **Figure. 4b** in the main text, and deleted **Figure. 4c** and removed **Figure. 4a** into supplementary. Meanwhile, we also deleted the redundant text, revised the remaining text accordingly.

Yes, we also ran the VarScan⁵ for tumor analysis. We use o2n-sam2sites and

VarScan to measure the mutation frequency separately after we mapped the CSs of o2n-seq libraries into the reference genome. Besides, we ran the VarScan to filter the strand bias mutations after identifying these mutations by 8X CSs. VarScan helped us a lot when we analysis o2n-seq data. VarScan could be further adapt to analysis our o2n-seq data, especially for high quality reads o2n-seq produced. VarScan and o2n-sam2sties, both of them were mentioned in the methods. Our previous expression on this was ambiguous, we revised the statements and highlighted in the “Methods” accordingly.

Why did the authors decide not to look at actual somatic mutations, i.e. what about somatic mutations shared between all three tumor samples but not the normal?

Response: Thanks for your comments. We detected lots of somatic mutations shared among all three tumors using o2n-seq, however, most of these mutations were high frequency mutations (MAF> 10%). These high frequency mutations might play important roles in the tumor evolution, however, they couldn't show the substantial advantage of o2n-seq, for they also can be detected by standard NGS method. Therefore, to better exhibit the substantial advantages of o2n-seq, we mainly focus on those low-frequency mutations.

There are a few differences between the o2nseq and varscan MAF (see supplementary tables). How can that be explained? Why is the difference highest for normal whereas o2nseq and varscan MAFs tumor samples look much more similar?

Response: We really appreciate your comments. The MAFs differences between o2n-sam2sites (script that used to measure MAFs in o2n-seq) and VarScan are due to the way to measure MAFs by o2n-sam2sites and VarScan is a little bit different. The way of MAFs calculation by VarScan as the following: CSs that used to produce pileup files only are filtered by mapping quality (>25) and base quality (60) before using to call variants and calculate the MAFs; In contrast, the way of MAFs calculation by o2n-sam2sites is: CSs that used to produce o2n-pileup are filtered not only by the mapping and base quality, but also by mismatch number (<2), INDELS et.al (see section “Mapping and variance calling” of “o2n-seq data processing”), only passed reads are used to calculate MAFs. The reason for the big differences between o2n-sam2sites and VarScan MAFs is because there are at least two variants existed in one read (around 100bp regions), and these variants are linkage variants. If one read covers two variants, this read will be discarded by o2n-sam2sites but not VarScan. So, the MAFs measured by o2n-sam2sites would be underestimated after discarding the reads with more than one variants. Excluded MAFs of those linkage variants, we found the MAFs of remaining variants measured by o2n-sam2sites to be highly concordant with measured by VarScan (**Figure. r4**). Moreover, this frequency difference pattern between o2n-sam2sites and VarScan provides a very efficient way to find the linkage variants and hotspots region in the genome. In the manuscript, to make sure the mutations were low-frequency mutations, we used “For low-frequency mutations, the

mutation minor allele frequencies calculated by o2n-sam2sites and VarScan should be lower than 0.1 concurrently.”

The differences between o2n-sam2sites and VarScan MAFs for normal and tumor are similar after filtering those linkage variants (**Figure. r4**).

Figure r4 The MAFs measured by o2n-sam2sites and VarScan after filtering linkage variants in N1, T1 and T2.

* Abstract: "O2n-seq reduces the error rate of NGS to less than 3×10^{-8} per nucleotide sequenced." This is a bit of an exaggeration, if I understand the results correctly. It's nowhere shown that it's "less" than that rate and in the discussion it's mentioned that the method has an error rate of 10^{-5} to 10^{-8} .

Response: Thanks for your question. As shown in the main text ("**Error rate and error pattern of o2n-seq**"), under different CSs (consensus sequences) support condition, the error rate of o2n-seq is different, from 1.18×10^{-5} (1X CSs) to 2.65×10^{-8} (5X CSs), and the error rate will decrease if more CSs (6X, 7X ...) are taken into account. However, because we didn't show the error rate of o2n-seq under 6X, 7X ..., we also think "O2n-seq reduces the error rate of NGS to 2.65×10^{-8} " is more appropriate than "O2n-seq reduces the error rate of NGS to less than 3×10^{-8} per nucleotide sequenced." Thanks, we revised and highlighted it in the abstract.

* The abstract doesn't make it clear that this is a new wetlab protocol as opposed to a new sensitive variant calling method.

Response: We really appreciate your suggestion. Our method is actually a new wetlab protocol. To distinguish it from drylab protocol, we revised the sentence "Here, we present o2n-seq, an ultrasensitive and high-efficiency method for discovering de novo, low-frequency mutations" into "Here, we present o2n-seq, an ultrasensitive and high-efficiency NGS library preparation method for discovering de novo, low-frequency mutations", and highlighted in the abstract.

* Introduction: one not referenced but related wetlab protocol is BaseSeq (PMID 25406369)

Response: Thanks for your helpful suggestion, we referenced this paper in the "**Introduction**", and highlighted it.

* Since the "genomic DNA is sheared into small pieces that are shorter than the length of a single PE read to ensure that each fragment can be sequenced twice independently in a pair of PE reads".

Could the authors show an example length distribution for the actually usable sequence stretch? Even given 2x 125 bp PE reads, after removing the adapters and merging the sequences the actually usable fragment must be rather small. Since effectively small, single-end fragments get mapped mapping uncertainty might be at the higher end for larger genomes (e.g. human).

Response: We really appreciate your comments. To make sure one pair of PE reads can sequence two different copies of original DNA fragment independently, the length of DNA fragment should be less than that of a single PE read. The length of DNA fragment is determined by the reads length of sequencing machine. For example, if we sequence the DNA using PE250 or PE300 modules, the length of DNA fragments can increase to 250bp or 300bp.

Figure. r2 showed the length distribution of actually usable fragment (Consensus sequences, CSs) of 2x 125 bp PE reads after removing the adapters and merging paired reads. Over 90% of CSs have length ranged from 80~120bp.

Yes, small, single-end fragments will lead to the increase of mapping error, especially for the large genomes (e.g. human). In our manuscript, to lower the impact of mapping error, we used strict filters to filter the alignments. Another strategy to reduce this impact is preparing and sequencing o2n-seq libraries using longer read length modules, such as PE250 or PE300. For more details, please refer to the responses for your second and third questions.

* HCC tumor application

"The sequencing results showed that 59.36% of the total data were successfully mapped to the target region and 99.92% of the total target region was covered. These results are comparable to those of standard NGS capture methods based on this target region panel (40.8% mapped to target region, 100% target region was covered)"

Could the authors provide a reference for this statement? The mappability seems rather low?

Response: We appreciate your comments. Because mappability of targeted capture approach is affected by many factors (kits provider, target size, et.al.)⁶⁻⁸, to explore the mapping rate of the targeted capture probe panel (0.42 Mb, customized from Roche NimbleGen) we had designed, we used this probe panel to capture one standard NGS library (not o2n-seq library) using normal human blood DNA as input. For this normal sample, we found 94.92% bases were successfully mapped to the human genome, but only 40.8% bases were mapped to this target region (covered 100% target region). To further confirm the result, we prepared another two standard NGS libraries using

normal human blood DNA, and captured with this probe panel. For these two libraries, 94.74%, 95.98% bases were mapped to human genome respectively, but, still, only 41.99%, 41.52% bases were successfully mapped to the target region respectively. In average, for these three standard NGS libraries captured with this target region, 95.21% ($\pm 0.67\%$) bases were successfully mapped to the human genome, only 41.44% ($\pm 0.59\%$) of total data were mapped to the target region. These standard NGS capture data clearly indicated the capture efficiency of o2n-seq were comparable with, even higher than that of standard NGS method. Thanks for pointing it out, we revised and highlighted the results in the main text.

* Bioinformatics: unfortunately no scripts or source code were shared. Having said this, the analysis seems simple enough.

Response: Thanks for your comment. The computational pipeline for processing o2n-seq data can be divided into five steps roughly, and the analysis isn't complicated. These five steps can be easily achieved by in-house scripts. But to make the analysis easier, we provided our in-house scripts of every step on:
<https://sourceforge.net/projects/o2n-seq/>

* Variant calling

Line 490: "SNPs were called using bcftools view 490 with the options of `-bvcg`". bcftools is the wrong choice of tools here: it assumes diploid organism. Better alternatives are VarScan (as used) and Lofreq or SNVer

Response: We really appreciate for your suggestion. As your suggestion, in revised main text we just used VarScan instead of using bcftools and VarScan to call SNPs in the section of "*E.coli* and *phix174* data processing". After that, we updated the different sites between *E. coli DH5 α* and *W3110* strains, high-confidence sites, and low-quality and variant sites. We recalculated the error rate and error pattern, reevaluated the efficiency of o2n-seq in the detection of low-frequency mutations based on those new sites sets. The new results are a little bit better than our previous results, eg. we found two more high-confidence sites which clearly distinguished the SNPs between the two *E.coli* strains, and both sites were successfully detected by o2n-seq in the *E.coli* 1:100 mix under 1X-5X criteria, so that the sensitivity of o2n-seq increased a little bit, while the FPR decreased a little bit. We have revised the results accordingly in the main text and methods. Thank you very much.

* Heterozygosity:

line 495 and table s1: "heterozygous" doesn't apply to *E.coli*. Might be better to call it "variant site" instead?

Response: Thanks for your kind suggestion, indeed, "variant site" was more accurate

in here, we modified the “heterozygous sites” into “variant sites” in line 495 and table s1, and highlighted in the methods of main text.

* Strand Bias:

line 513: "The identified mutations were further filtered by Varscan --min-var-freq 0.001 to filter strand bias mutations." That's an MAF filter, not for strand bias filter. But out of interest: did the authors look at strand-bias? This artifact seems to affect many variant calling methods for largely unknown reasons, which are thought to be PCR related. They occur mostly in low-frequency mutation at high coverage. Such filter is often used (e.g. in samtools) to remove false positive and could potentially help here as well, if it is present at all.

Response: We appreciate your comments. Yes, we actually had looked at and filtered the strand bias. We also noted the strand bias could affect the variant calling. We indeed filtered the strand bias mutations by **VarScan --min-var-freq 0.001 --strand-filter 1**. There was a typo in previous manuscript, and we fixed this mistake and highlighted in the methods of revised manuscript, thanks.

References for the responses to reviewer #2

1. Kennedy, S.R. *et al.* Detecting ultralow-frequency mutations by Duplex Sequencing. *Nature Protocols* **9**, 2586-606 (2014).
2. Lou, D.I. *et al.* High-throughput DNA sequencing errors are reduced by orders of magnitude using circle sequencing. *Proceedings of the National Academy of Sciences of the United States of America* **110**, 19872–19877 (2013).
3. Wang, K. *et al.* Ultra-precise detection of mutations by droplet-based amplification of circularized DNA. *BMC Genomics* **17**, 1-12 (2016).
4. Wang, K. *et al.* Using ultra-sensitive next generation sequencing to dissect DNA damage-induced mutagenesis. *Scientific Reports* **6**, 25310 (2016).
5. Koboldt, D.C. *et al.* VarScan: variant detection in massively parallel sequencing of individual and pooled samples. *Bioinformatics* **25**, 2283-2285 (2009).
6. Samorodnitsky, E. *et al.* Comparison of Custom Capture for Targeted Next-Generation DNA Sequencing. *The Journal of Molecular Diagnostics* **17**, 64-75 (2015).
7. Schmitt, M.W. *et al.* Sequencing small genomic targets with high efficiency and extreme accuracy. *Nat Methods* **12**, 423-5 (2015).
8. Chung, J. *et al.* The minimal amount of starting DNA for Agilent’s hybrid capture-based targeted massively parallel sequencing. *Scientific Reports* **6**, 26732 (2016).

Reviewers' Comments:

Reviewer #1 (Remarks to the Author)

The authors have made an appropriate revision of the manuscript in response to the points I raised. I appreciate the points raised by reviewer 2, and I think the manuscript has improved considerably after taking the two reviews into account, and may now be acceptable for publication.

Reviewer #2 (Remarks to the Author)

The authors put a lot of work into the rebuttal and addressed all issues I think. While doing so, the manuscript became unfortunately even denser, but I don't see a simple way around this.

Regarding the error rate: I still prefer the more honest (and still impressive) wording in the discussion ("o2n-seq exhibits an error rate of 10⁻⁵-10⁻⁸") over the more generic and inflated one in the abstract ("o2n-seq reduces the error rate of NGS to 2.65 × 10⁻⁸")...

REVIEWERS' COMMENTS:

Reviewer #1 (Remarks to the Author):

The authors have made an appropriate revision of the manuscript in response to the points I raised. I appreciate the points raised by reviewer 2, and I think the manuscript has improved considerably after taking the two reviews into account, and may now be acceptable for publication.

Response: We really appreciate your previous comments and suggestions, which helped us to improve our method and manuscript a lot.

Reviewer #2 (Remarks to the Author):

The authors put a lot of work into the rebuttal and addressed all issues I think. While doing so, the manuscript became unfortunately even denser, but I don't see a simple way around this.

Regarding the error rate: I still prefer the more honest (and still impressive) wording in the discussion ("o2n-seq exhibits an error rate of 10^{-5} - 10^{-8} ") over the more generic and inflated one in the abstract ("o2n-seq reduces the error rate of NGS to 2.65×10^{-8} ")...

Response: Thanks for your kind suggestion, we also prefer the more honest wording. We revised "o2n-seq reduces the error rate of NGS to 2.65×10^{-8} " into "o2n-seq reduces the error rate of NGS to 10^{-5} - 10^{-8} " and highlighted in the abstract of the main text. We really appreciate your effort in reviewing this manuscript.